



# Closure of In-Situ Measured Aerosol Backscattering and Extinction Coefficients with Lidar Accounting for Relative Humidity

Sebastian Düsing[1], Albert Ansmann[1], Holger Baars[1], Joel C. Corbin[3,4], Cyrielle Denjean[1,2], Martin Gysel-Beer[3], Thomas Müller[1], Laurent Poulain[1], Holger Siebert[1], Gerald Spindler[1], Thomas Tuch[1], Birgit Wehner[1], Alfred Wiedensohler[1]

[1]Leibniz Institute for Tropospheric Research, 04318 Leipzig, Germany.
[2]now at: CNRM, Université de Toulouse, Météo-France, CNRS, Toulouse, France.
[3]Laboratory of Atmospheric Chemistry, Paul Scherrer Institute, 5232 Villigen PSI, Switzerland
[4]now at: Metrology Research Centre, 1200 Montreal Road, National Research Council Canada, Ottawa, ON K1A 0R6, Canada.

*Correspondence to*: Sebastian Düsing (duesing@tropos.de)

**Abstract.** Aerosol particles contribute to the climate forcing through their optical properties. Measuring these aerosol optical properties is still challenging, especially considering the hygroscopic growth of aerosol particles, which alters their optical properties. Lidar and in-situ techniques can derive a variety of aerosol optical properties, like aerosol particle light extinction, backscattering, and absorption. But these techniques are subject to some limitations and uncertainties. Within this study, we compared with Mie-theory modeled aerosol optical properties with direct measurements. At dried state they were with airborne and ground-based in-situ measurements; at ambient state with lidar estimates. Also, we examined the dependence of the aerosol particle light extinction-to-backscatter ratio, also lidar ratio (LR), to relative humidity. The used model was fed with measured physicochemical aerosol properties and ambient atmospheric conditions. The model considered aerosol particles in an internal core-shell mixing state with constant volume fractions of the aerosol components over the entire observed aerosol particle size-range. The underlying set of measurements was conducted near the measurement site Melpitz, Germany, during two campaigns in summer, 2015, and winter, 2017, and represent Central European background aerosol conditions. Two airborne payloads deployed on a helicopter and a balloon provided measurements of microphysical and aerosol optical properties and were complemented by the polarization Raman lidar system Polly$^{XT}$ as well as by a holistic set of microphysical, chemical and optical aerosol measurements derived at ground level. The calculated aerosol optical aerosol properties agreed within 13% (3%) with the ground-based in-situ measured aerosol optical properties at a dried state (relative humidity below 40%) in terms of scattering at 450 nm wavelength during the winter (summer) campaign. The model also represented the aerosol particle light absorption at 637 nm within 8% (18%) during the winter (summer) campaign and agreed within 13% with the airborne in-situ aerosol particle light extinction measurements during summer. During winter, in a comparatively clean case with equivalent black carbon mass-concentrations of around 0.2 µg m$^{-3}$ the modeled airborne measurement-based aerosol particle light absorption, was up to 32-37% larger than the measured values during a relatively clean period. However, during a high polluted case, with an equivalent black carbon mass concentration of around 4 µg m$^{-3}$, the modeled aerosol particle light absorption coefficient was, depending on the wavelength, 13-32% lower than the measured values. Spread and magnitude of the disagreement highlighted the importance of the aerosol mixing state used within the model, the requirement of the inclusion of brown carbon, and a wavelength-dependent complex refractive index of black and brown carbon when such kind of model is used to validate aerosol particle light absorption coefficient estimates of, e.g., lidar systems.

Besides dried state comparisons, ambient modeled aerosol particle light extinction, as well as aerosol particle light backscattering, were compared with lidar estimates of these measures. During summer, on average, for four of the twelve conducted measurement flights, the model calculated lower aerosol particle light extinction (up to 29% lower) as well as backscattering (up to 32% lower) than derived with the lidar. In winter, the modeled aerosol particle light extinction coefficient was 17%-41% lower, the aerosol particle light backscattering coefficient 14%-42% lower than the lidar estimates.



For both, the winter and summer cases, the Mie-model estimated reasonable extinction-to-backscatter ratios (LR). Measurement-based Mie-modeling showed evidence of the dependence of the LR on relative humidity (*RH*). With this result,

we presented a fit for lidar wavelengths of 355, 532, and 1064 nm with an underlying equation of $f_{LR}(RH, \gamma(\lambda)) = f_{LR}(RH = 0, \lambda) \times (1 - RH)^{-\gamma(\lambda)}$ and estimates of $\gamma(355 \text{ nm}) = 0.29 \ (\pm 0.01)$, $\gamma(532 \text{ nm}) = 0.48 \ (\pm 0.01)$, and $\gamma(1064 \text{ nm}) = 0.31 \ (\pm 0.01)$. However, further measurements are required to entangle the behavior of the LR with respect to different aerosol types, to set up a climatology, and to assess the influence of the aerosol mixing state.

        This comprehensive study combining airborne and ground-based in-situ and remote sensing measurements, which

simulated multiple aerosol optical coefficients in the ambient and dry state, is with its complexity unique of its kind.



## 1 Introduction

Aerosol particles interact with incoming solar radiation and alter its pathway through the atmosphere by scattering and absorption. The wavelength-dependent aerosol particle light scattering ($\sigma_{sca}(\lambda)$) and absorption coefficients ($\sigma_{abs}(\lambda)$) are

measures of these interactions and the sum of both is described by the aerosol particle light extinction coefficient ($\sigma_{ext}(\lambda)$). The effect of aerosol particles on the radiative budget of the atmosphere, including fast atmospheric adjustments is known as effective radiative forcing and is estimated to -0.45 Wm$^{-2}$ within an uncertainty range of -0.95 to +0.05 Wm$^{-2}$ (Boucher et al., 2013). These uncertainties are not fully understood yet and demand further research.

Direct in-situ aerosol measurements with unmanned aerial vehicles (UAV; Altstätter et al., 2018), helicopter-borne

payloads, e.g., with the Airborne Cloud and Turbulence Observations System (ACTOS; e.g., Siebert et al., 2006, Ditas et al., 2012, Wehner et al., 2015; Düsing et al., 2018), tethered-balloon payloads (e.g., Ferrero et al., 2019, Brunamonti et al., 2020), and zeppelins (e.g., Rosati et al., 2016a) provide insights on the role of the aerosol in atmospheric processes. Remote sensing techniques such as light detection and ranging (lidar) profile the atmosphere concerning the aerosol particle optical properties via measuring the aerosol particle light backscattering (Weitkamp, 2005). Lidar and in-situ measurements, when used to

constrain climate models, are useful tools to help to decrease the uncertainties of the radiative forcing estimates. Raman-lidar systems, such as Polly$^{XT}$ (Engelmann et al., 2016), can measure the aerosol particle light extinction and backscattering at several wavelengths $\lambda$. However, during the daytime, the estimates of aerosol particle light extinction underlie uncertainties due to the background skylight interfering with the Raman-signals. A variety of lidar systems do exists and many cannot filter out these background signals. Hence, the directly measured wavelength-dependent aerosol particle light backscattering

coefficient ($\sigma_{bsc}(\lambda)$) is often transformed into $\sigma_{ext}(\lambda)$ with the so-called aerosol-type and wavelength dependent lidar ratio, $LR(\lambda)$. This parameter describes the aerosol particle light extinction-to-backscatter ratio and is, e.g., related to the wavelength of incoming light, the shape of the aerosol particles, the aerosol particle number size distribution (PNSD) and aerosol chemical composition. $LR(\lambda)$ estimates during daytime have been derived via a combination of direct lidar $\sigma_{bsc}(\lambda)$ and columnar sun-photometer measurements (Guerrero-Rascado et al., 2011). A sun-photometer derives the columnar integral of $\sigma_{ext}(\lambda)$, the

aerosol optical depth (AOD). An effective columnar $LR(\lambda)$ then can be estimated by minimizing the difference between measured AOD and the integrated lidar-based $\sigma_{ext}(\lambda)$ derived with a $LR(\lambda)$. When the Klett-Fernald method (Klett, 1982, Fernald et al., 1972) is used to derive $\sigma_{ext}(\lambda)$ and $\sigma_{bsc}(\lambda)$ with lidar, the $LR(\lambda)$ is kept height-constant and introduces uncertainties, e.g., because these columnar $LR(\lambda)$ do not represent layers of different aerosol types within the atmosphere and can deviate from in-situ observations (Guerrero-Rascado et al., 2011).

Direct in-situ aerosol measurements, as well as the modeling of optical aerosol coefficients, are useful to cross-validate remote sensing techniques like lidar and vice versa. For instance, lidar-based $\sigma_{bsc}(\lambda)$ have been compared with balloon-borne in-situ measurements (Brunamonti et al., 2020) and Mie-modeling (Ferrero et al., 2019). Airborne in-situ aerosol measurements, also, provide vertically resolved insights into aerosol optical properties when deployed on airborne platforms (e.g., Rosati et al., 2016a, Düsing et al., 2018, Tian et al., 2020).

However, airborne in-situ measurements are usually conducted under controlled dried conditions. Lidar on the other hand examines the aerosol under ambient conditions. Previous studies have shown the dependence of $\sigma_{ext}(\lambda)$ (Skupin et al., 2013 and Zieger et al., 2013), and $\sigma_{bsc}(\lambda)$ (Haarig et al., 2017) on ambient RH. Navas-Guzmán et la. (2019) utilized these effects to investigate the aerosol hygroscopicity with lidar. $LR(\lambda)$ is based on the RH-dependent $\sigma_{bsc}(\lambda)$ and $\sigma_{ext}(\lambda)$ and calculations by Sugimoto et al. (2015) indicated that $LR(\lambda)$ is RH-dependent as well. Ackermann (1998) provided a numerical

study based on pre-defined aerosol types with distinct size-distribution shapes to establish a power series to describe the $LR(\lambda)$ in terms of RH. Salemink et al. (1984) have shown a linear relationship between the $LR(\lambda)$ and the RH. Therefore, the effect of the RH must be considered when comparing in-situ measurements and modeling approaches with remote-sensing techniques. Both studies show an inconclusive dependence of the $LR(\lambda)$ to the RH with different representations (linear, power series), showing that further research is still needed. Also, a quantification based on direct in-situ measurements is still missing.



Based on selected cases, this study presents the results of two field-experiments conducted in June 2015, and Winter, 2017 at the regional Central-European background measurement facility in Melpitz, located in the East of Germany. In both, a combination of airborne in-situ and remote sensing measurements, accompanied by a sophisticated set of ground-based in-situ measurements, were conducted under different atmospheric conditions and aerosol load. This study aims at first to compare remote sensing measurements of $\sigma_{bsc}(\lambda)$ and $\sigma_{ext}(\lambda)$ with calculated airborne in-situ measurement-based modeled coefficients,

utilizing a closure study. Second, it gives insights on the *LR* enhancement, and answers the question to which extent the lidar ratio depends on the ambient *RH* at three different wavelengths based on in-situ measurement-based optical modeling under the given aerosol conditions at the measurement site. Third, the study evaluates the capability of the used Mie-model to recreate measured $\sigma_{abs}(\lambda)$ at different wavelengths to create a tool for the validation lidar-based $\sigma_{abs}(\lambda)$ estimates as shown by Tsekeri et al. (2018). This study, which includes simultaneous modeling of $\sigma_{bsc}(\lambda)$, $\sigma_{ext}(\lambda)$, and $\sigma_{abs}(\lambda)$ in ambient and dried state based

on ground-based and vertical resolved in-situ, and remote-sensing measurements, is unique in its complexity.

This work is structured as follows. First, an overview of the measurements site and the deployed instrumentations is given. Afterwards, details about the used optical model including a description of the applied input parameters as well as the validation with in-situ reference instrumentation are given. Subsequently, the comparison of Mie-modeled and measured aerosol optical properties is presented and discussed separated into the summer and winter experiment. This also includes with

a short overview of the meteorological and aerosol conditions during the experiments. The quantification of the lidar ratio enhancement with respect to *RH* is given for the summer case. Finally, conclusions are formulated based on the results.


## 2 Experiments

In this study, the data assembled during two campaigns near Melpitz, Saxony, Germany, are examined. The first campaign named "Melpitz Column" or *MelCol-summer*, unless otherwise stated ongoing referred to as summer campaign, was conducted in May and June 2015 with an intensive measurement period including ground-based and air-borne in-situ measurements between June 13 and June 28. The second campaign, *MelCol-winter*, took place in February and March 2017, and thus is referred to as the winter campaign in the further course of this paper. The upcoming sections give an overview of the conducted experiments, introduce the Melpitz Observatory with its characteristic features, and provide an overview of the applied instrumentation on the ground as well as in the air.

### 2.1 Melpitz Observatory

Both campaigns took place at the central European background station at Melpitz, Saxony, Germany. Melpitz Observatory (51° 31' N, 12° 55' E; 84 m a.s.l.) is located in Eastern Germany in a rural, agriculturally used area 44 km northeast of Leipzig. About 400 km to the north is the Baltic Sea, and about 1000 km to the west is the Atlantic Ocean. Detailed information about Melpitz Observatory is given in Spindler et al. (2010, 2013). As part of various measurement networks, such as GUAN (German Ultra-fine Aerosol Network; Birmili et al., 2016), ACTRIS (Aerosols, Clouds and Trace gases Research Infrastructure), and GAW (Global Atmosphere Watch), and the measurement facility LACROS (Leipzig Aerosol and Cloud Remote Observations System; Bühl et al., 2013) Melpitz Observatory comprises comprehensive instrumentation in quasi-continuous operation, for high-quality, long-term observations and can be adapted to the needs as required. An overview of the continuously operating instrumentation is presented in the following. Details about specific instrumentation additionally added during the campaigns will be given within respective subsections.

#### 2.1.1 Ground in-situ instrumentation

In both campaigns, the PNSD was measured by a combination of a Dual Mobility Particle Size Spectrometer (D-MPSS, TROPOS-type; Birmili et al., 1999) with 10% accuracy and Aerodynamic Particle Size Spectrometer (APSS, mod. 3321, TSI Inc., Shoreview, MN, USA) with 10%-30% uncertainty depending on the size-range (Pfeifer et al., 2016).

A D-MPSS consist of a bipolar diffusion charger, two differential mobility analyzer (DMA; Knutson and Whitby, 1975) and two condensation particle counters (CPC; mod. 3010 and UCPC; mod. 3776, TSI Inc., Shoreview, MN, USA). The bipolar charger transforms the aerosol into a well-defined charge equilibrium, according to Fuchs (1968) and Wiedensohler et al. (1988). The TROPOS-type DMAs selects the charged aerosol particles concerning their electrical mobility, and the CPC then counts their number concentration. Overall this setup covers an aerosol particle size range of 3-800 nm in mobility diameter ($D_\mathrm{m}$). The PNSD is available every 20 minutes, and a scan duration is ten minutes. The final D-MPSS PNSD used in this study was derived utilizing an inversion routine (Pfeifer et al., 2014) accounting for multiple charged aerosol particles, including a diffusion loss correction based on the method of "equivalent pipe length" (Wiedensohler et al., 2012).

For the calculation of the optical properties with the Mie-theory, spherical particles must be assumed. Therefore, we assumed that all aerosol particles measured by the D-MPSS system used here are spherical, and the $D_\mathrm{m}$ is equal to the volume equivalent diameter ($D_\mathrm{v}$). The quality of the PNSD measurements is assured by frequent calibrations as described in Wiedensohler et al. (2018). To cover the entire size-range from 10 nm to 10 µm, the APSS PNSD extended the D-MPSS PNSD. For this purpose, the aerodynamic diameter ($D_\mathrm{aer}$) of the APSS was converted into $D_\mathrm{v}$ applying:

$$D_\mathrm{v} = \sqrt{\frac{\chi \times \rho_0}{\rho_\mathrm{aer}}}\, D_\mathrm{aer} = \sqrt{\frac{\rho_0}{\rho_\mathrm{eff}}}\, D_\mathrm{aer}, \text{ with} \tag{1}$$

$$\frac{\rho_\mathrm{aer}}{\chi} = \rho_\mathrm{eff}, \tag{2}$$





following DeCarlo et al. (2004). Thereby $\rho_0$ corresponds to the standard density of 1 g cm$^{-3}$, $\rho_{aer}$ to the aerosol density, $\rho_{eff}$ to the effective aerosol density of 1.5 g cm$^{-3}$ for fine mode aerosol and already accounts for the shape of the larger aerosol particles expressed with the shape factor $\chi$. The effective density of 1.5 g cm$^{-3}$ was chosen, because with that a best overlap of the APSS and T-MPSS PNSD was achieved for the majority of merged PNSDs. Also, this effective density fits reasonably well to the

findings of Tuch et al. (2000) and Poulain et al. (2014) with reported aerosol particle densities of $1.53 \pm 0.31$ g cm$^{-3}$ and 1.4 g cm$^{-3}$ to 1.6 g cm$^{-3}$, respectively. Although shape factor and aerosol particle density are usually size-dependent, we assumed a constant density and shape of the aerosol particles for all the measurements of the APSS. At visible wavelengths, the coarse-mode of the PNSD is less efficient than the fine-mode in terms of aerosol particle light scattering and extinction. Hence, for aerosols dominated by accumulation mode particles, the underlying assumption is appropriate to calculate the

extinction and scattering properties of the aerosol.

In addition to these continuously running instruments at Melpitz Observatory, a Quadrupole Aerosol Chemical Speciation Monitor (Q-ACSM, Aerodyne Res. Inc, Billerica, MA., USA; Ng et al., 2011) measured the mass concentration of non-refractory particulate matter (PM). Ammonium (NH$_4$), sulfate (SO$_4$), nitrate (NO$_3$), and chlorine (Cl), as well as the organic aerosol mass, have been derived in the fine-mode regime (NR-PM$_1$). Further details on the Q-ACSM measurements

at Melpitz can be found in Poulain et al., (2020). An ion-pairing scheme (ISORROPIA II; Fountoukis and Nenes, 2007) was utilized to derive the chemical compounds of the aerosol particles at 293 K and 0% *RH*. Furthermore, a DIGITEL DHA-80 (Walter Riemer Messtechnik e.K., Hausen/Röhn, Germany) high volume aerosol sampler collected daily the PM$_{10}$ (10 denotes an aerodynamic diameter of the aerosol particles of 10 µm) aerosol particles on a quartz-fiber filter (Type MK 360, Munktell, Grycksbo, Sweden) with a total flow of 30 m$^3$ h$^{-1}$. Among others, Müller (1999), Gnauk et al. (2005), and Herrmann et al.

(2006) provide detailed information about the aerosol sampler. The sampled quartz-fiber filter was analyzed offline and allowed the determination of the total aerosol particle mass concentration (in this study we focus on PM$_{10}$), water-soluble ions, and the mass of elemental carbon (EC). The EC mass concentration ($m_{EC}$) was measured following the EUSAAR2 protocol (Cavalli et al., 2010),

A continuously operating Multi-Angle Absorption Photometer (MAAP; Model 5012, Thermo Scientific, Waltham,

MA, USA; Petzold and Schönlinner, 2004) recorded the $\sigma_{abs}(\lambda)$ at Melpitz Observatory at a wavelength of 637 nm with an uncertainty of 10% (Müller et al., 2011) to 12% (Lack et al. 2014). Several corrections were applied to the aerosol particle light absorption measurements of the MAAP. Following Müller et al. (2011), a wavelength correction factor of 1.05 was applied to all MAAP-data in this study. Furthermore, observations conducted in Melpitz by Spindler et al. (2013) and Poulain et al. (2014) have shown that the submicron aerosol regime contains 90% of the total PM$_{10}$ equivalent black carbon (eBC;

Petzold et al., 2013) mass concentration ($m_{eBC}$). Hence, on the estimated $m_{eBC}$ data, a correction factor of 0.9 was applied to match the corresponding PM$_1$ measurements of the Q-ACSM. With $m_{EC}$) and these absorption measurements, $m_{eBC}$was derived using a time-dependent ($t$) mass absorption cross-section related to the MAAP wavelength of 637 nm ($MAC(t, \lambda = 637$ nm)) with:

$$m_{\text{eBC}}(t, 637\text{nm}) = \frac{\sigma_{\text{abs}}(t(\text{hourly}),637\text{nm})}{MAC(t(\text{daily}),637\text{nm})}. \tag{3}$$

The daily average $MAC(t,$ 637 nm$)$ was derived by dividing the daily $m_{EC}$ by the daily (midnight to midnight) mean of the measured $\sigma_{abs}(637$ nm):

$$MAC(t(daily),637\text{nm}) = \frac{m_{\text{EC,Digitel}}(t(\text{daily}))}{\sigma_{\text{abs,MAAP}}(t(\text{daily}),637\text{nm}))}. \tag{4}$$

Following this approach, a mean daily $MAC(637$ nm$)$ of 10.4 m$^2$ g$^{-1}$ (median 10.9 m$^2$ g$^{-1}$; IQR: 7.1 to 12.3 m$^2$ g$^{-1}$) was derived for the period between February 1 and March 15, 2017. Recently, Yuan et al. (2020) provided $MAC(870$ nm$)$ estimates for the

winter campaign period of this study of 7.4 m$^2$ g$^{-1}$ (geometric mean value, range from 7.2 to 7.9 m$^2$ g$^{-1}$) which relates to a $MAC(637$ nm$)$ of around 10.8 m$^2$ g$^{-1}$ (10.5 to 11.5 m$^2$ g$^{-1}$) assuming an absorption Ångström exponent (*AAE*) of 1.2 (taken



from Yuan et al., 2020). Zanatta et al. (2016), also, reported a geometric mean $MAC$(637 nm) of 8.2 m$^2$ g$^{-1}$ (geometric standard deviation of 1.5 m$^2$ g$^{-1}$). For the period between June 1 and June 30, 2015, a mean daily $MAC$(637 nm) of 7.3 m$^2$ g$^{-1}$ (median 7.2 m$^2$ g$^{-1}$; IQR: 6.0 to 8.4 m$^2$ g$^{-1}$) was estimated at Melpitz Observatory, which agrees with the 7.4 m$^2$ g$^{-1}$ previously reported

by Nordmann et al. (2013) and is slightly lower than the geometric mean $MAC$(637 nm) of 9.5 m$^2$ g$^{-1}$(geometric standard deviation of 1.38 m$^2$ g$^{-1}$) reported by Zanatta et al (2016) for the aerosol at Melpitz during summer. However, the estimates of Nordmann et al. (2013) were derived with Raman spectroscopy. Hence, the here estimated $MAC$(637 nm*)* values for summer and winter seem reasonable as well, but will be evaluated in depth later on. The specific volume fractions of each aerosol compound, $f_{v,i}$, were derived based on the Q-ACSM and MAAP measurements dividing the mass of each aerosol compound

with its respective density. Appendixtable 1 lists the density of each derived aerosol compound. Moteki et al. (2010) reported that it is accurate within 5% to assume the density of non-graphitic carbon at 1.8 g cm$^{-3}$. Therefore, in this study a BC density of 1.8 g cm$^{-1}$ is used.

These measurements were completed by a Nephelometer (mod. 3563, TSI Inc., Shoreview, MN, USA), which measures the $\sigma_{sca}(\lambda)$ at 450, 550, and 700 nm with a relative uncertainty by calibration and truncation of about 10% (Müller et

al., 2009). The error of the Nephelometer measurements due to truncation and illumination was corrected following Anderson and Ogren. (1998).

The aerosol particle hygroscopicity parameter $\kappa$, introduced by Petters and Kreidenweis (2007), represents a quantitative measure of the aerosols water uptake characteristics and depends on the chemical composition of the aerosol particles as well as their size. A Volatility Hygroscopicity-Tandem Differential Mobility Analyser (VH-TDMA), first introduced by Liu et al.

(1978), measures the hygroscopic growth, and hence water uptake, of aerosol particles at a specific $RH$. This instrument was deployed at Melpitz Observatory during the summer campaign. The VH-TDMA measured the hygroscopic growth of aerosol particles in six different size-bins (30, 50, 75, 110, 165, and 265 nm) from which the size-resolved aerosol hygroscopicity $\kappa(D_p)$ was inferred. The scientific community uses a variety of VH-TDMAs, but detailed insights on the system deployed here provide Augustin-Bauditz et al. (2016). The inferred $\kappa(D_p)$ allows to extrapolate the hygroscopic growth of aerosol particles

to another $RH$. For the calculation of the hygroscopic growth of the aerosol particles under ambient conditions, we assumed $\kappa(D_p)$ for diameters smaller 30 nm is equal to $\kappa$(30 nm) and for diameters larger 265 nm is equal to $\kappa$(265 nm). During the winter campaign, no size-resolved direct hygroscopicity measurements were available. Therefore, the hygroscopicity of the aerosol particles encountered in the winter campaign was derived based on the parallel conducted measurements of the aerosol chemical composition utilizing the Zdanovskii, Stokes, and Robinson (ZSR; Zdanovskii, 1948; Stokes and Robinson, 1966)

volume-weighted mixing rule considering the hygroscopicity parameter of every single aerosol compound $\kappa_i$ listed in Appendixtable 1. A comparison of the size-segregated $\kappa(D_p)$ estimates of the VH-TDMA with bulk Q-ACSM measurements during the summer campaign has shown a 1:1 agreement with high correlation ($R^2 = 0.98$, fit through origin) at 165 nm. Hence, bulk Q-ACSM measurements represent the aerosol at a size of around 165 nm. However, the bulk Q-ACSM approach might over- or underestimates the hygroscopicity of aerosol particles lower or larger than 165 nm. Furthermore, Düsing et al. (2018)

have conducted an optical closure experiment comparing Mie-based aerosol particle light extinction and backscatter coefficients with lidar measurements, using both, $\kappa$ estimates based on chemical composition and cloud condensation nuclei counter measurements at 0.2% supersaturation. In the case of the chemical composition measurements the agreement with the lidar was within 10% in terms of the aerosol particle light extinction coefficient. Hence, using $\kappa$ from the bulk Q-ACSM measurements is a feasible approach.

**2.1.2 Ground-based remote sensing**

In addition to the in-situ measurements on the ground, in both campaigns a Lidar system was used to determine $\sigma_{bsc}(\lambda)$ and $\sigma_{ext}(\lambda)$. This system was Polly$^{XT}$, a 3+2+1 wavelengths Raman polarization lidar system, in the first version introduced by Althausen et al. (2009). The Polly$^{XT}$ version in this study was introduced by Engelmann et al. (2016) and did operate with



three channels for aerosol particle light backscattering and two for aerosol particle light extinction. During the summer
campaign a near-field channel at 532 nm was available. After the summer campaign, Polly$^{XT}$ was updated and equipped with
an additional near-field channel at 355 nm and therefore available during the winter-campaign. Vertical profiles of these
aerosol properties were available each 30 s with a vertical resolution 7.5 m. The geometry of emitted laser and far field-of-
view (FOV) leads to a partial overlap below 800 m altitude, which is known as the overlap height and can be determined
experimentally (see Wandinger and Ansmann, 2002). Below 800 m, an overlap correction was applied to the lidar data (see
Engelmann, 2016, and Wandinger and Ansmann, 2002). The standard far FOV is 1 mrad and the near FOV is 2.2 mrad
(Engelmann et al., 2016). The automated data evaluation routines and quality check control are presented in detail in Baars et
al. (2016). An intercomparison campaign presented by Wandinger et al. (2016) including different EARLINET (European
Aerosol Research LIdar NETwork) instruments, including the system within this study (see Lidar system named le02 therein)
has shown a maximum deviation of less than 10%. Hence, we assume a 10% measurement uncertainty of the $\sigma_{bsc}(\lambda)$
measurements.

During daytime, the signal-to-noise ratio in the Raman-channels is too weak due to solar radiation to provide robust
Raman $\sigma_{ext}(\lambda)$. Therefore, in this and other studies, the $\sigma_{bsc}(\lambda)$ have been converted to $\sigma_{ext}(\lambda)$ by means of the extinction-to-
backscatter ratio, also known as lidar ratio (*LR*, in sr), with:

$$\sigma_{ext} = \sigma_{bsc} \times LR. \tag{5}$$

*LR* is an aerosol intensive property.

In the past, several studies, investigated the *LR* of different aerosol types with ground-based lidar systems (Haarig et
al., 2016, Mattis et al., 2004, Wang et al., 2016, and Ansmann et al., 2010; with an airborne lidar system by Groß et al. (2013).
Cattrall et al. (2005) estimated *LR*s at 550 nm and 1020 nm wavelength based on retrievals of direct sky radiance and solar
transmittance measurements. Tao et al. (2008) and Lu et al. (2011) determined the *LR* with a synergistic approach combining
space-borne and ground-based lidar. Düsing et al. (2018) provide *LR* based on airborne in-situ measurements estimated with
Mie-theory. All these investigations clearly show that the *LR* is highly dependent on the predominant aerosol types. Müller et
al. (2007) and Mattis et al. (2004) provided an overview of the *LR* for different aerosol types. Mattis et al. (2004) provided
long-term (2000-2003) estimates of the *LR* for central European haze (anthropogenic aerosol particles) of 58 (±12) sr for 355
nm, 53 (±11) sr for 532 nm, and 45 (±15) sr for 1064 nm wavelength, respectively. In this study, the measured $\sigma_{bsc}(\lambda)$ was
transformed into $\sigma_{ext}(\lambda)$ with these estimates. The uncertainties of the estimates of Mattis et al. (2004) and the measurements
uncertainties of the lidar system were accounted in the derived $\sigma_{ext}(\lambda)$.

Additionally, a sky spectral radiometer (mod. CE318, Cimel Electronique, 75011 Paris, France) was deployed during
both intensive periods of both campaigns as part of the AERONET observations. This pointed sun radiometer derived the
AOD at several wavelengths, and Holben et al. (1998) provide detailed insights on the working principle of this instrument. It
was used to cross-check the lidar retrievals in terms of validation of the integrated $\sigma_{ext}(\lambda)$ profiles with the AERONET AOD.

With a combination of both, the lidar and the sun-photometer, profiles of $\sigma_{abs}(\lambda)$ can be estimated using the
Generalized Aerosol Retrieval from Radiometer and Lidar Combined data algorithm (GARRLiC; Lopatin et al., 2013). But
AOD at 404 nm of 0.4 and more are needed for this purpose, thus we could not apply it for our study.

### 2.1.3 Airborne in-situ measurements during summer

**The Airborne Cloud and Turbulence Observation System**

During the intensive period of the summer campaign, a set of state-of-the-art instruments, installed on the airborne
platform ACTOS (Siebert et al., 2006), determined microphysical and aerosol optical properties. ACTOS was designed as an
external cargo under a helicopter with a 150 m long aerial rope and was operated maximum ascend and descend speeds of
6 m s$^{-1}$. Ambient *RH* and temperature (*T*) were recorded as well and were averaged to a temporal resolution of 1 Hz. A data





link was established between ACTOS and a receiver station installed on the helicopter and allowed the scientist on board of the helicopter a real-time data observation to adjust flight height and track.

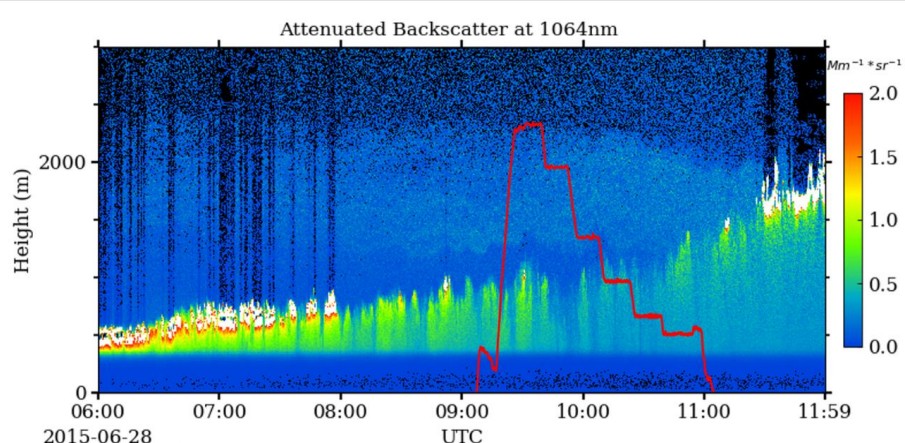

**Figure 1: Attenuated aerosol particle light backscatter coefficient ($\sigma_{bsc}(\lambda)$, color coded) measured with Polly$^{XT}$ lidar at 1064 nm on June 28, 2015, between 6:00 UTC and 12:00 UTC. White colors indicate values larger than 2.0 Mm$^{-1}$ sr$^{-1}$. The red line represents the**
**flight pattern of ACTOS in terms of altitude in m above ground.**

A typical flight pattern of one of the conducted measurement flights is displayed exemplarily for June 28, 2015, in Figure 1 as a red line. Typically, a measurement flight lasted around two hours and started with a profile to characterize the atmosphere vertically up to altitudes of 2700 m and to identify atmospheric layers of interest. Afterwards, sections of constant
flight height, so-called "legs" were flown with at least 10 minutes duration to realize measurements within on altitude level and to increase the counting statistics for other measurements, such as the PNSD with a lower time resolution and such as the aerosol particle absorption coefficient deployed on ACTOS. Figure 1, also, displays color-coded the attenuated $\sigma_{bsc}(\lambda)$ at 1064 nm in Mm$^{-1}$ sr$^{-1}$ measured by Polly$^{XT}$ lidar on June 28, 2015, between 06:00 UTC and 12:00 UTC. Bright white color represents a strong backscatter signal and indicate clouds. The development of the planetary boundary layer is visible with the
increasing cloud bottom height of 500 m at 06:00 UTC and around 1600 m altitude at 12:00 UTC. Also, the residual layer containing some aerosol layer aloft the top of the planetary boundary layer (PBL) between 1250 m and 2300 m is visible indicated by greenish colors. The payload, therefore, was sampling in the free troposphere as well as within the planetary boundary layer and was sampling different aerosol populations. A short period at around 09:30 UTC of low-level clouds interfered the measurements of the lidar during the flight.

**Aerosol sampling on ACTOS**

On ACTOS, a custom-made silica-bead based diffusion dryer dried the air sample to ensure an aerosol humidity below 40% following the recommendations of Wiedensohler et al. (2012). A TROPOS-built MPSS determined the PNSD with a temporal resolution of two minutes covering a size range of 8 nm to 230 nm. This temporal resolution translates into a vertical spatial resolution of several 100 m depending on the ascent/descent speed of the helicopter. Like the D-MPSS on the ground,
this MPSS included a bipolar charger (here mod. 3077A, TSI Inc., Shoreview, MN, USA) containing radioactive Kr-85, a TROPOS-type DMA (Hauke-type, short) and a condensation particle counter (CPC; mod. 3762A, TSI Inc., Shoreview, MN, USA) with a lower cut-off diameter ($D_{p,50\%}$; the CPC detects 50% of the aerosol particles with this diameter) of around 8 nm and counting accuracy of 10%. An optical particle size spectrometer (OPSS; here mod. skyOPC 1.129, GRIMM Grimm Aerosol Technik, Ainring, Germany) recorded the optical equivalent PNSD covering an aerosol particle size range of 350 nm

to 2.8 µm (optical diameter) with a temporal resolution of 1 Hz. The corresponding two-minute averaged OPSS PNSD extended the MPSS PNSD. The detailed geometry of the optical cell inside the instrument is unknown. Hence, a correction regarding the complex aerosol refractive index ($n = n_r + in_i$) could not be applied to the data set. The upper cut-off of the inlet system is estimated at around 2 µm following Kulkarni et al. (2011). The PNSD has been corrected concerning aspirational and diffusional losses following Kulkarni et al. (2011) and Wiedensohler et al. (2012) using the method of the "equivalent pipe
length".

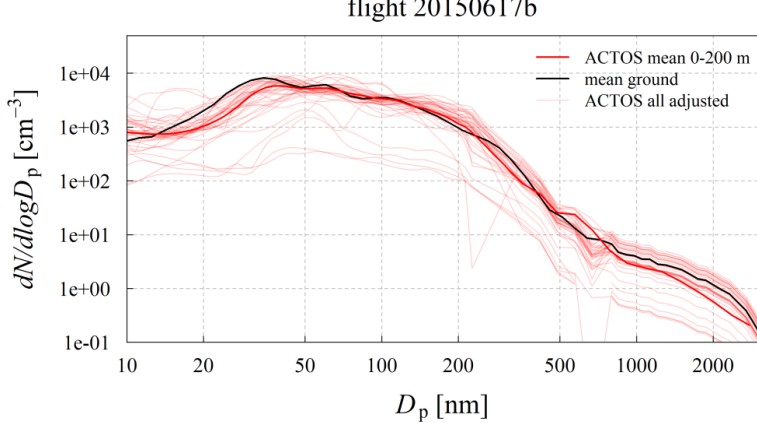

**Figure 2: PNSD at dried state derived during flight 20150617b. The red line indicates the mean PNSD in the atmospheric layer between 0 – 200 m sampled with the ACTOS MPSS and OPSS. The black line represents the mean PNSD derived on the ground during the ACTOS flight time. Red transparent thin lines display the PNSDs derived with ACTOS adjusted with the height-**
**corrected PNSD measured at Melpitz Observatory.**

     The quality of the airborne in-situ measurements was checked by comparing average of the lowermost 200 m with the ground in-situ measurements (see Figure 2). The intercomparisons revealed a distinct underestimation of the aerosol particle number concentration above 800 nm in optical diameter (see Figure 2). This underestimation is caused presumably
due to a mixture of losses within the system which cannot be addressed appropriately and the here missing refractive index correction of the OPSS which would shift the OPSS PNSD more to larger particle diameters (see Alas et al., 2019). Since, the in-situ instrumentation at ground is quality-assured, the ground-based measurements is the reference and was utilized to correct the airborne measurements. Therefore, above 800 nm, the airborne in-situ PNSD recorded by the OPSS was replaced and extended with a height-corrected PNSD measured on the ground at Melpitz Observatory establishing a non-fixed altitude-
correction factor $f_h$. The altitude-correction factor $f_h(h,$ scan) was calculated according Eq. (6):

$$f_h(h,\text{scan}) = \frac{N_{\text{OPSS}}(h)}{N_{\text{OPSS},<200\text{m}}}, \qquad\qquad\qquad (6)$$

Where $N_{\text{OPSS},<200\,m}$ is the mean aerosol number concentration derived with the OPSS in the lowermost 200 m. $N_{\text{OPSS}}(h,$ scan) is the mean aerosol particle number concentration detected by the OPSS during the corresponding scan-time of the MPSS at a given altitude $h$ ($N_{\text{OPSS}}(h)$). Advantageously, this method accounts for uncertainties introduced due to differences in the
complex refractive index of the calibration aerosol and the prevalent aerosol and accounts for the upper cut-off limit of the inlet-system.

     Furthermore, aerosol optical properties were measured onboard ACTOS. The Single Channel Tri-Colour Absorption Photometer (STAP; Brechtel Manufacturing Inc., Hayward, CA, USA) derived $\sigma_{\text{abs}}(\lambda)$ at 450, 525, and 624 nm wavelength, respectively. Briefly, the STAP evaluates $\sigma_{\text{abs}}(\lambda)$ based on light attenuation measurements behind two filters with a spot-size
of around 1.75 x $10^{-5}$ m$^{-2}$. In this study quartz-fiber filter (Pallflex membrane filters, type E70-2075W, Pall Corp., Port Washington, NY, USA) were used. On one filter, the aerosol matters deposits, and one filter spot stays clean downstream the



first filter. A photodetector detects the intensity of light of the given wavelength behind these filter spots. All raw data have been recorded on a 1 Hz time-resolution. At default the STAP estimates $\sigma_{abs}(\lambda)$ based on 60 s running averages of the measured intensities. At this averaging period, the measurement uncertainty is estimated to 0.2 Mm$^{-1}$. Based on differential light
attenuation measurements between two time-steps, the STAP calculates the $\sigma_{abs}(\lambda)$. Filter-loading and the enhancement of absorption due to multiple-scattering within the filter-material have been corrected following Ogren (2010) and Bond et al. (1999). These corrections include the real-time estimated filter-transmission dependent loading correction factor:

$$f(\tau) = (1.0796\tau + 0.71)^{-1}, \tag{7}$$

where the transmission $\tau$ is defined as the ratio of the intensity $I(t)$ measured at time $t$ and the blank-filter intensity $I_0 = I(t_0)$.
Due to the limited computational power of the internal chip onboard of the STAP $\sigma_{ext}(\lambda)$ was recalculated at 1 Hz time resolution during the postprocessing with larger precision. Since the STAP was in an early developing state we faced issues concerning the implemented analog-to-digital converter and data of the STAP sampled during summer is not presented in this study.

Additionally, a Cavity Attenuation Phase Shift Monitor (CAPS PM$_{ssa}$; Aerodyne Research, Billerica, MA, USA) was
measuring $\sigma_{ext}(\lambda)$ and $\sigma_{sca}(\lambda)$ at 630 nm wavelength each second. The truncation error of $\sigma_{sca}(630\ nm)$was not corrected; therefore, within this study, we focus on $\sigma_{ext}(630\ nm)$estimated with a 5% accuracy.

**2.1.4 Airborne in-situ measurements during winter**

During MelCol-Winter, the tethered balloon system BELUGA (Balloon-bornE modular Utility for profilinG the lower Atmosphere, Egerer et al., 2019) carried a set of payloads, which determined meteorological conditions, including ambient $T$
and $RH$, as well as microphysical and aerosol optical properties. The 90 m³ helium-filled balloon was attached on a 2 km long tether (3 mm Dyneema®), an electric winch allowed profiling with a climb and sink-rate of 1 to 3 m s$^{-1}$.

A temperature-insolated container included the same STAP also deployed during the summer campaigndetermined $\sigma_{abs}(\lambda)$. An OPSS (mod. 3330, TSI Inc., Shoreview, MN, USA) was sampling the PNSD in a range of 0.3 to 10 µm in 16 size bins every 10 seconds. The OPSS PNSD was corrected in terms of the complex aerosol refractive index. Here, a complex
aerosol refractive index of 1.54 + i0 was used since this resulted in OPSS PNSD with a good overlap to the MPSS PNSD. The imaginary part of the complex aerosol refractive index was forced to 0 because it leads to a significant overestimation of the coarse mode in the PNSD when the imaginary part of the complex aerosol refractive index is above 0 (see Alas et al., 2019). Note, that this complex aerosol refractive index is not the refractive index used in the Mie model.

The missing size-range of the PNSD, here all particles smaller than 0.3 µm in optical diameter, was extended with
the altitude-corrected average ground-based PNSD of the corresponding flight period analogue to the summer campaign. Here, the variable altitude correction factor $f_h$ from Eq.(6) was for each OPSS PNSD the ratio of the aerosol particle number concentration detected by OPSS within the lowermost 50 m ($N_{OPSS,<50\ m}$) and the aerosol particle number concentration detected by the OPSS at an altitude $h$ ($N_{OPSS}(h)$). Particles larger than 800 nm have not been replaced by the PNSD measurements at ground since the refractive index correction was applied to the OPSS data.
Varying wind-speeds during the campaign changed the inclination of the aerosol inlet accordingly. Therefore, we do not account for the varying upper cut-off of the inlet. However, calculations following Kulkarni et al. (2010) with an inclination angle of 90° show that 50% of 10 µm aerosol particles with a density of 2 g cm$^{-3}$ are aspirated by the inlet at a wind-speed of around 0.8 m s$^{-1}$. Diffusional losses at the OPSS size-range are negligible. The aerosol was dried with a silica-bead based dryer similar to the one on ACTOS to dampen sudden changes in the $RH$ of the aerosol stream, which can have significant influences
on the filter-based absorption measurements of the STAP as shown, for instance, by Düsing et al. (2019). They estimated a deviation of around 10.08 (± 0.12) Mm$^{-1}$ %$^{-1}$ s (10.08 Mm$^{-1}$ per unit change of $RH$ (in %) per second), which is significant, especially under clean conditions. An $RH$ sensor (model HYT939, B+B Thermo-Technik GmbH, Donaueschingen, Germany) sensor recorded the $RH$ of the sampled air downstream of the drier.



### 3 Modeling of aerosol optical properties

Mie's theory (Mie, 1908) allows calculating the optical properties of aerosol particles under the assumption that these particles are spherical. The Mie-model applied here fulfilled three major tasks. First, it was tested to what extent it can reproduce measured $\sigma_{abs}(\lambda)$ with the given constraints. Second, it was compared to lidar-based $\sigma_{bsc}(\lambda)$ and $\sigma_{ext}(\lambda)$ based on airborne in-situ measurements accounting the ambient $RH$. Third, it derived $LR(\lambda)$ at ambient aerosol conditions to examine the $LR$-$RH$ dependence.

For both campaigns, an adapted, Mie-model, written in Python 2.6 (package PyMieSca v1.7.5; Sumlin et al., 2018), simulated the aerosol optical properties; in particular, $\sigma_{bsc}(\lambda)$, $\sigma_{ext}(\lambda)$, $\sigma_{sca}(\lambda)$, and $\sigma_{abs}(\lambda)$ for eight different wavelengths. From $\sigma_{bsc}(\lambda)$ and $\sigma_{ext}(\lambda)$ the Mie-based $LR(\lambda)$ ($LR_{Mie}(\lambda)$) was derived. For slightly non-spherical particles, Mie-theory is still applicable to particles with a size-parameter $x = \pi\,D_p\,\lambda^{-1}$ of less than five; for particles with a larger $x$, Mie-theory results in a lower $LR(\lambda)$ than the slightly non-spherical particles would have (Pinnick et al., 1976). At 355 nm, for instance, Mie-theory would

underestimate the $LR(\lambda)$ already for non-spherical particle with a diameter larger than 570 nm, the corresponding thresholds for 532 nm and 1064 nm are 850 nm and 1700 nm. Also, giant particles, usually non-spherical, result in a larger $LR(\lambda)$ than calculated with Mie-theory.

Regarding the mixing state of the aerosol, three different approaches are considered in the scientific community: 1) external mixture, in which each aerosol compound is represented by its own PNSD, 2) internally homogeneous mixture, with

homogeneously mixed aerosol compounds within the aerosol particles, and 3) the internal core-shell mixture, in which a core of a specific compound, like sea salt or light-absorbing carbon, is surrounded by a shell of, e.g., organics or inorganic salts. Regarding internally mixed aerosols, Ma et al. (2012) have shown that for the aged aerosol conditions at Melpitz, the core-shell mixing model usually is the better representation of the internally mixed approaches to estimate the aerosol optical properties. Rose et al. (2006) furthermore have shown that the number fraction of externally mixed soot aerosol particles at

80 nm diameter is rather low in Melpitz, indicating a majority of internally mixed aerosol particles at this size-range. The study of Yuan et al. (2020), conducted at Melpitz observatory, has shown coating thicknesses of several tens of nm of BC cores with a diameter of about 200 nm estimated for February 2017. Based on these findings, the core-shell internal mixture model was utilized in this study to calculate the aerosol optical properties for both campaigns. We assumed that the aerosol particles consist of a non-water-soluble core of light-absorbing carbon and a shell of water-soluble, non-absorbing material. However,

it must be mentioned that in general the mixing of aerosol particles is rather complex and a more sophisticated approach would be to consider mixtures of aerosol particle populations. For instance, a mixture could be a combination of homogeneously mixed aerosol particles containing no BC, and aerosol particles containing a light absorbing BC core surrounded by a shell of inorganic salts, organic material, or something else. However, the number fraction of both populations would remain unclear.

Similar to Düsing et al. (2018), the Mie-model used the aerosol particle diameter and number concentration, extracted

from the dried-state PNSD, the aerosol particle core diameter, and the complex refractive index of the aerosol particle core and shell as input parameters to derive the aerosol particle optical properties in the dried state. The aerosol particle core diameter $D_c$ was calculated with:

$$D_c = D_p \times f_{v,eBC}^{\frac{1}{3}}, \tag{8}$$

where $f_{v,eBC}$ is the volume fraction of eBC and was assumed to be constant over the entire size-range. The volume fraction of

the eBC particles was estimated as described in Section 2.1.1. Due to a lack of airborne chemical composition measurements, we assumed that the chemical composition derived on ground was representative for the planetary boundary layer in both campaigns.

Within the model, an additional optional module calculated the aerosol optical properties in the ambient state. This module required additional information about the aerosol and environment, like its hygroscopicity parameter $\kappa$, and the ambient

temperature $T$ and $RH$. At first, the module simulated the hygroscopic growth of the aerosol particles utilizing the semi-





empirical parameterization of Petters and Kreidenweis (2007). For this, the in Sect. 2.1.1 introduced $\kappa$-estimates from the ground in-situ measurements were utilized. In a second step, it estimated the volume fraction of water of each aerosol particle based on these hygroscopic growth simulations.

Following Ma et al. (2014) and references therein, the complex refractive index of water-soluble compounds was set to be 1.53 + 1e-6i, with a 0.5% uncertainty of the real part and 0% of the imaginary part, respectively. The water-insoluble light-absorbing (eBC) compounds were estimated to have a wavelength-independent complex refractive index of 1.75 + 0.55i, with a 4% and 6.6% uncertainty, respectively. This approach leads to inaccuracies especially for calculating $\sigma_{abs}(\lambda)$ since the complex aerosol refractive index depends on the wavelength. Bond and Bergstrom (2006), e.g., recommended a complex refractive index of BC at 550 nm of 1.95 + 0.79i at 550 nm whereas Moteki et al. (2010) reported values of 2.26 + 1.26i at
1064 nm.

Also, only BC was considered, whereas brown carbon (BrC), usually organic material and hence part of the particle shell, was not. But, BrC is especially effective in light absorption at lower wavelengths, whereas the contribution of BC to $\sigma_{abs}(\lambda)$ decreases towards lower wavelengths. A brief discussion of the spectrally resolved Mie-based $\sigma_{abs}(\lambda)$ follows in Sect. 4.2.1.

Hale and Querry (1973) provided the complex refractive index of water (liquid; 25°C). Following this publication, the mean (± standard deviation) of the real part of the complex refractive index of water is 1.33 (± 0.0043) in the range from 0.3 to 1.0 µm wavelength. The imaginary part is negligibly small (4.5e-7) in this wavelength range. Hence, the complex refractive index of water was set to 1.33 + 0i with an assumed real part uncertainty of 0.5%. At ambient state, the complex refractive index of the aerosol particle shell was derived based on the volume weighted ZSR mixing rule of the complex
refractive index of the water-soluble components and the additionally added water. Although the sampled aerosol was dried, it always contained a small amount of residual water, which is negligible for the hygroscopic growth calculations. In the Mie-model, each estimate of the aerosol optical properties was derived with a Monte-Carlo approach with n = 50 runs. Bevor each run, the input parameters were varied according to their uncertainty with a Gaussian normal-distribution or an uniform-distribution when the Gaussian normal-distribution creates physically unreasonable input parameters, e.g., a negative volume
fraction of eBC, or negative ambient $RH$. Table 1 summarizes the input parameters of the Mie-model with the uncertainties and the underlying distribution for the variation within the Monte-Carlo approach.

**Table 1: Overview of the input parameters of the Mie-model, the corresponding assumed uncertainties, and the underlying type of distribution for the variation of the input parameter.**

| parameter | uncertainty | underlying distribution for the model |
|---|---|---|
| $dN/d\log D_p(D_p)$ | 10% | uniform |
| $D_p$ | 0% | - |
| $n_{eBC}$ | 4% real part; 6% imaginary part | normal |
| $n_{water}$ | 0.5%; - | normal |
| $n_{sol}$ | 0.5%; - | normal |
| $RH$ | standard deviation of the mean (scan period) | uniform |
| $T$ | standard deviation of the mean (scan period) | uniform |
| $f_{v,eBC}$; $f_{v,sol}$ | standard deviation of mean (flight period) | uniform |
| $\kappa(D_p)$ H-TDMA summer | standard deviation of the mean (day) | uniform |
| $\kappa$ bulk Q-ACSM winter | standard deviation of the mean (flight period) | uniform |


The underlying assumptions within the Mie-Model were validated using a correlation of the measured and Mie-based aerosol optical properties in the dried state (see Figure 3), and with the in-situ measured $\sigma_{ext}(630\ nm)$ derived on ACTOS with

the CAPS (see Figure 4). Considering the correlation with the ground-based in-situ measurements of $\sigma_{sca}(450\ nm)$, the model agrees within 3% during the summer campaign (underestimation, Figure 3a)) and within 13% (overestimation, Figure 3b))

during the winter period. Based on the correlation in Figure 3, the Mie-model reproduced the $\sigma_{abs}(\lambda)$ derived with the MAAP at 637 nm within 8% (Figure 3b)) during winter, and within 18% (Figure 3a)) during the summer period overestimating the measured $\sigma_{abs}(\lambda)$ in both cases.

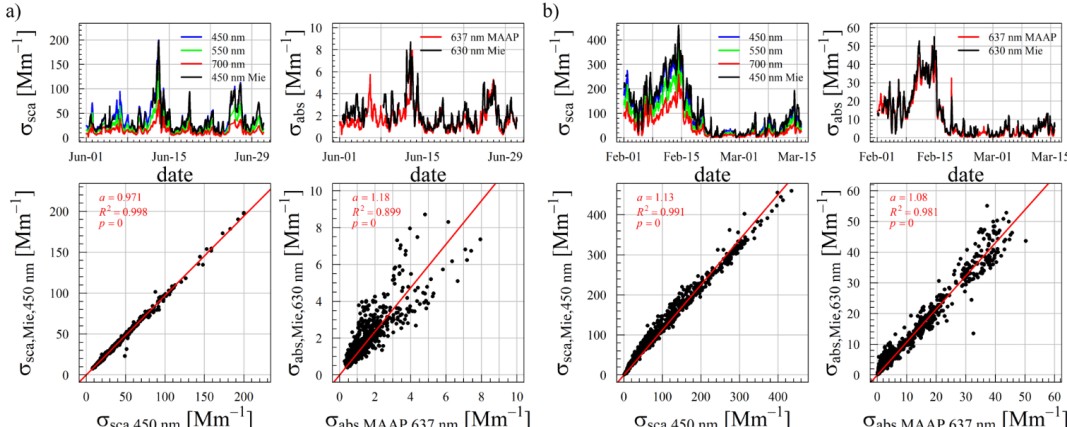

**Figure 3: Time series (upper panels) and scatter plot (lower panels) of modeled and measured aerosol particle light scattering ($\sigma_{sca}(\lambda)$, left panels) and absorption ($\sigma_{abs}(\lambda)$, right panels) coefficients derived with the Mie-model and the Nephelometer and MAAP for different wavelength (color coded)at Melpitz Observatory during the summer (a) and winter campaign (b).**

In the summer case, two distinct clusters in the $\sigma_{abs}(\lambda)$, one above and one below the fitting line, indicating different aerosol types and that the model constraints might represented the prevalent aerosol type of lower cluster better since the data

points are close the 1:1 line. The aerosol represented by the lower cluster was prevalent at Melpitz from 13 June 2015 on and the comparison of the modeled and measured $\sigma_{ext}(\lambda)$ ($\sigma_{sca}(\lambda)$) has shown an agreement within 4% (2%). Therefore, the mixing approach within the model is a good representation of the aerosol the intensive period of the measurement campaign in summer between 15 June and 28 June 2015.

However, the model utilized rough assumptions to represent the aerosol. Besides the assumption of a wavelength-

independent complex aerosol refractive index, the assumption of a constant volume fraction of eBC resulted in an underestimation of the BC content in the smaller aerosol particles and led to an overestimation in the larger aerosol particles, because BC usually is largely found in the aerosol accumulation and Aitken mode (Bond et al., 2013) with a mass peak at around 250 nm of BC core diameter. Also, the coating thickness of same-sized soot cores is not constant and the size of BC cores covers only a certain size-range as shown by Ditas et al. (2018). No size-resolved BC mass concentration measurements

have been available during the summer campaign, and would also be limited to a certain size-range. Therefore, the implementation of a constant eBC volume fraction within an optical model is a handy approach and is often used in other studies (e.g., Düsing et al., 2018 and Ma et al., 2014, 2012).

Furthermore, the model validation in terms of absorption is based on the *MAC*(637 nm) estimates based on the MAAP measurements and hence most representative at this wavelength. Modeled $\sigma_{abs}(\lambda)$ at lower or larger wavelengths could deviate

from measurements because of a different value of *MAC*($\lambda$).





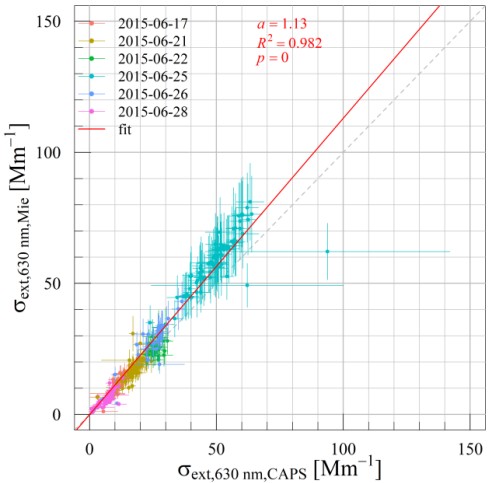

**Figure 4: Scatter plot of the in-situ airborne measurement-based aerosol particle light extinction coefficient derived with the CAPS and the Mie model at 630 nm in the dried state. The red line indicates the linear fit of both, the gray dashed line represents the 1:1 line, and color-coded are the measurement days of the summer campaign.**


However, considering the airborne in-situ correlation, the model agrees to measured $\sigma_{ext}$(630 nm) within 13% (slope = 1.13 with $R^2$ = 0.98; $p$ = 0) averaged over all available data points of all conducted flights. But, the modeled $\sigma_{ext}$(630 nm) overestimate the measured one especially on June 25 (light blue data points). Excluding that day from the correlation, the model would overestimate the measured $\sigma_{ext}$(630 nm) by 2.2% ($R^2$ = 0.98), which is within the measurement

uncertainty of the CAPS. Note that for the airborne in-situ correlation, the underlying airborne PNSD used in the Mie-model was not corrected for diffusional and aspirational loss, because both systems were sampling through the same inlet system. In winter, the altitude corrected PNSD measured at ground which was used to replace of the missing aerosol particle size range (up to 300 nm) was, however, corrected for the diffusional losses inside the tubing. Diffusional losses inside the tubing of the balloon platform lower the in-situ measured $\sigma_{abs}$($\lambda$). Therefore, the in-situ measured $\sigma_{abs}$($\lambda$) would have been smaller than

modeled ones by default. To which extent, however, remains unclear.

Nevertheless, the agreement of both approaches, Mie modeling and in-situ measurements, at ground and airborne implies that the model constraints provide a good representation of the "real" aerosol properties, at least in the dried state with the limitation of a $MAC$(637 nm) applied to all considered wavelengths.



## 4 Results

### 4.1 MelCol-summer

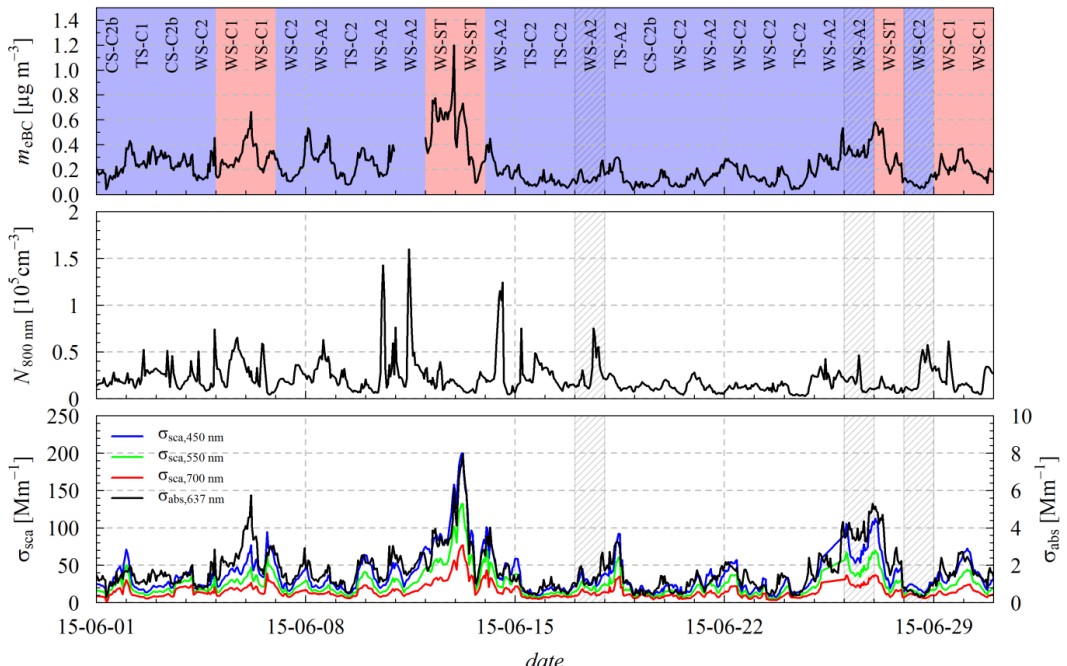

**Figure 5:** Upper panel shows the equivalent black carbon mass concentration ($m_{eBC}$) from June 1 to June 30, 2015. Color codes represent clean (blue) and polluted (red) trajectory clusters with the given keys for each day following the trajectory clustering in Sun et al. (2020) and is explained below. The second panel shows the total number concentration of all aerosol particles between 5 and 800 nm in diameter ($N_{800nm}$), and the lower panel displays the aerosol particle light scattering ($\sigma_{sca}$) and absorption coefficient ($\sigma_{abs}$). Grey shaded areas show the measurement days investigated in more detail.

Figure 5 shows in the top panel the time series of $m_{eBC}$ at Melpitz Observatory during June, 2015, derived with the daily (midnight to midnight) filter-measurement based $MAC$(637 nm) estimates. Each day in both campaigns was assigned to its corresponding air-mass back-trajectory from a pool of in total 15 clusters following Sun et al. (2020). These clusters were assigned by the season (cold season, CS; transition season, TS; and warm season, WS), and the prevalent synoptic pattern. The abbreviation ST indicates a stagnant pattern, A indicates anticyclonic patterns with air masses originating in eastern (1) and western (2) Europe. C represents a cyclonic pattern with air masses originating from the south (1) and north (2). The prevalent trajectory cluster is assigned with red or blue colors indicating polluted or clean conditions and the respective key. Clusters with keys CS-ST, CS-A1, CS-A2, CS-C1, TS-A1, WS-ST, WS-A1, and WS-C1 represent polluted conditions. Briefly, the clustering is based on a k-means clustering method for meteorological back-trajectories (Dorlin et al., 1992). Further details present Ma et al. (2014) and the supplementary material of Sun et al. (2020).

The middle panel displays the total aerosol particle number concentration of particles up to 800 nm in diameter. The $\sigma_{sca}(\lambda)$ and $\sigma_{abs}(\lambda)$ at 450, 550, and 700 nm, and 637 nm, respectively, are shown in the bottom panel. During this period, the average $m_{eBC}$ was 0.23 ($\pm$ 0.14) µg m$^{-3}$ (range from 0.04 to 1.2 µg m$^{-3}$), which is in the range of the median $m_{eBC}$ for cleaner air-masses (cluster keys: CS-C2a, CS-C2b, TS-A2, TS-C1, TS-C2, WS-A2, and WAS-C2) as reported by Sun et al. (2020).

During the summer campaign, 14 flights were conducted, two of which were test flights. Low-level clouds strongly biasing the lidar-measurements. Therefore, after screening the weather conditions of all conducted flights for periods of low-



level cloud coverage, four measurement flights performed on three days have been left for further investigation with preferable

mostly clear sky conditions. The gray shaded boxes in Figure 5 mark the three investigated days without low-level clouds of this study. The three investigated days cover a wide range of the observed $m_{eBC}$ (0.03 to 0.58 µg m$^{-3}$) during the intensive period between June 15 and June 28, 2015. Daily mean $m_{eBC}$ of 0.14 (± 0.05) µg m$^{-3}$ were observed during June 17, 0.35 (± 0.05) µg m$^{-3}$ during June 26, and 0.095(± 0.03) µg m$^{-3}$ during June 28, 2015. The three days are characterized by westerly inflows (trajectory cluster WS), and the airmass originated from the North Atlantic (WS-A2 (clean); June 17 and June

26 and WS-C2 (clean); June 28; Sun et al., 2019). In the following, two flights, flight b on June 17 and flight a on June 26, 2015, and their corresponding atmospheric profiles will be investigated in depth. Flight 20150617b was conducted at relatively clean conditions, whereas flight 20150626a was conducted within a period of comparatively high $m_{eBC}$. The comparison of the modeled and measured optical properties for all days will be shown in Table 2.

### 4.1.1 Model vs. Lidar

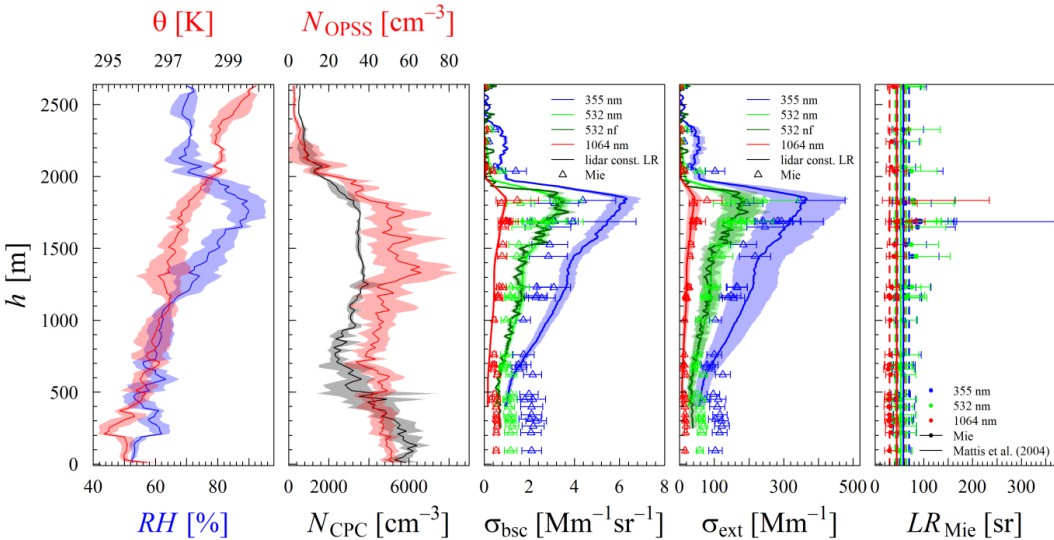


**Figure 6:** Vertical profiles of the 20m-layer averages of the ambient $RH$ (blue, first panel), potential temperature $\theta$ (red, first panel), the aerosol particle number concentration of all particles ($N_{CPC}$; black, second panel), and of the particles detected by the OPSS ($N_{OPSS}$; red, second panel). The third panel displays the measured (colored lines, averaged from 08:35 to 09:00 UTC) and modeled

(triangles, each PNSD scan on ACTOS) aerosol particle light backscattering coefficient ($\sigma_{bsc}(\lambda)$) for the given wavelengths 355 nm (blue), 532 nm (green), and 1064 nm (red). The forth panel the aerosol particle light extinction coefficient ($\sigma_{ext}(\lambda)$), correspondingly. The last panel shows the modeled extinction-to-backscatter ratio ($LR_{Mie}(\lambda)$) derived with the Mie-model (dots) at the respective wavelengths (colored dots), and the vertical lines represent the $LR$ given in Mattis et al. (2004) with the given uncertainty estimates with dashed lines. Uncertainty-bars around the Mie-based $\sigma_{bsc}(\lambda)$ and $\sigma_{ext}(\lambda)$ denote range within ± three times standard deviation; around $LR_{Mie}(\lambda)$ they denote the range of possible $LR_{Mie}(\lambda)$ resulting from the uncertainties of the modeled $\sigma_{bsc}(\lambda)$ and $\sigma_{ext}(\lambda)$. The

given profiles were derived during the flight b between 08:08 and 09:58 UTC on June 26, 2015.

Figure 6 shows the vertically resolved atmospheric conditions during the measurement flight conducted between 08:08 and 09:58 UTC on June 26, 2015. The 20 m-layer averages of microphysical aerosol particle properties, the ambient $RH$ and $T$, and the measured (average between 08:35 and 09:00 UTC) and modeled aerosol optical properties of each PNSD scan

are shown. The top of the PBL is about at an altitude of around 2 km. From 2000 m to 0 m altitude, the total aerosol particle number concentration, measured by the CPC ($N_{CPC}$), as well as the number concentration for aerosol particles larger than 350 nm ($N_{OPSS}$) indicate two different aerosol layers. Between 1200 and 1800 m altitude, a layer with constant $N_{CPC}$ of around 4000 cm$^{-3}$ and $N_{OPSS}$ of around 55 cm$^{-3}$ has been observed. In the layer from 700 m to 0 m altitude, $N_{CPC}$ steadily increased towards the ground up to 5000 cm$^{-3}$, while $N_{OPSS}$ scatters around 45 cm$^{-3}$. For this layer, the model calculated larger optical





coefficients then observed with the lidar. Above 700 m altitude, the model calculated lower $\sigma_{bsc}(\lambda)$ at 355 nm and 532 nm and slightly lower $\sigma_{ext}(355\ nm)$. That indicates different aerosol populations in these layers. For the layer near ground (up to 600 m altitude) the assumptions within the model might led to aerosol optical coefficients larger than the lidar-based ones.

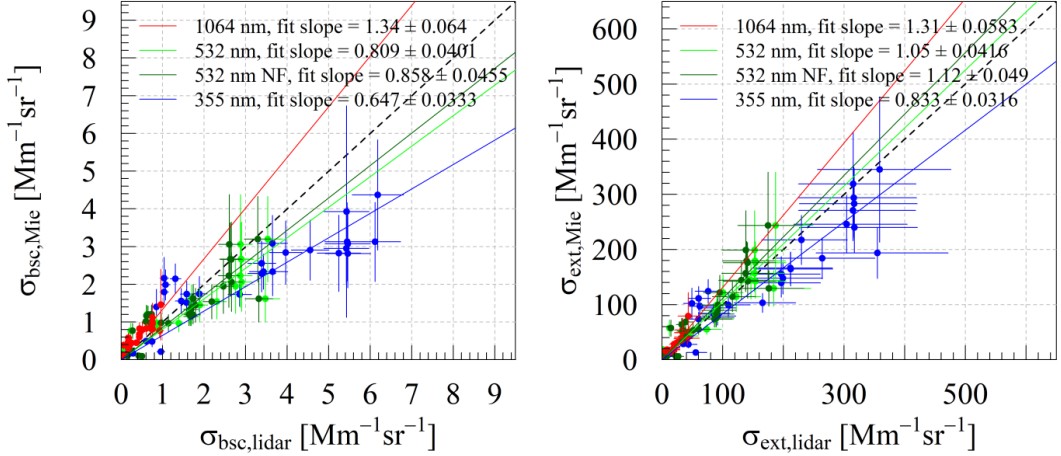

**Figure 7: Scatter plots of the measured (lidar) and modeled (Mie) ambient state aerosol particle light backscattering ($\sigma_{bsc}(\lambda)$) and extinction ($\sigma_{ext}(\lambda)$) coefficient derived during flight 20150626a. Vertical uncertainty-bars indicate the range within ±three times standard deviation of the mean. Horizontal uncertainty-bars denote the uncertainty of the lidar estimates. Colored lines represent linear fit at the corresponding color for 1064 nm (red), 532 nm (green, NF dark green), and 355 nm (blue). Black dashed line represents the 1:1 line.**


       Figure 7 summarizes the results shown in third and fourth panel Figure 6. Regarding $\sigma_{bsc}(\lambda)$, the Mie-model calculated around 34 (±6.4)% larger values than measured with the lidar at 1064 nm wavelength, 19.1 (±4)% lower values at 532 nm, and 35.3 (±3.3)% lower values at 355 nm. Considering $\sigma_{ext}(\lambda)$, the estimates of the Mie-model were 31 (±5.8)% larger than the lidar-based estimates at 1064 nm wavelength and by 5 (±4)% larger at 532 nm. At 355 nm the Mie-model calculated around

16.7 (±3)% lower aerosol particle light extinction coefficients than derived with the lidar.

       Panel 5 of Figure 6 displays the spectrally resolved modeled $LR_{Mie}(\lambda)$ and the $LR(\lambda)$ with the given uncertainty-range reported by Mattis et al. (2004). In the lowermost 1200 m $LR_{Mie}(\lambda)$ was relatively constant and the $RH$ did increase from ground to 1200 m from around 50% to 70%. The impact of the $RH$ on the $LR(\lambda)$ was small due to small hygroscopic growth of the aerosol particles in this $RH$ range. Under these conditions, the mean $LR_{Mie}(\lambda)$ was 54 sr at 355 nm and 532 nm, respectively.

This mean $LR_{Mie}(\lambda)$ is in the range of reported $LR(\lambda)$ for urban haze aerosol reported by Müller et al. (2007) and Mattis et al. (2004) and is reasonable considering also the $LR(532\ nm)$ of polluted dust aerosol of 60 sr reported by Omar et al. (2009. The anthropogenic influence (urban, polluted) is indicated by a larger $m_{eBC}$ compared to June 17, and 28 (see Figure 5). The mean $LR_{Mie}(1064\ nm)$ below 1200 m altitude was 30 sr and agrees with the findings of Omar et al. (2009). They reported a $LR(1064\ nm)$ of 30 sr based on satellite-borne lidar observations for clean continental, polluted continental, and polluted dust

aerosol. Above 1200 m altitude the $LR_{Mie}(\lambda)$ followed the trend of the $RH$ up to the PBL top indicating a $LR$-$RH$ dependence.





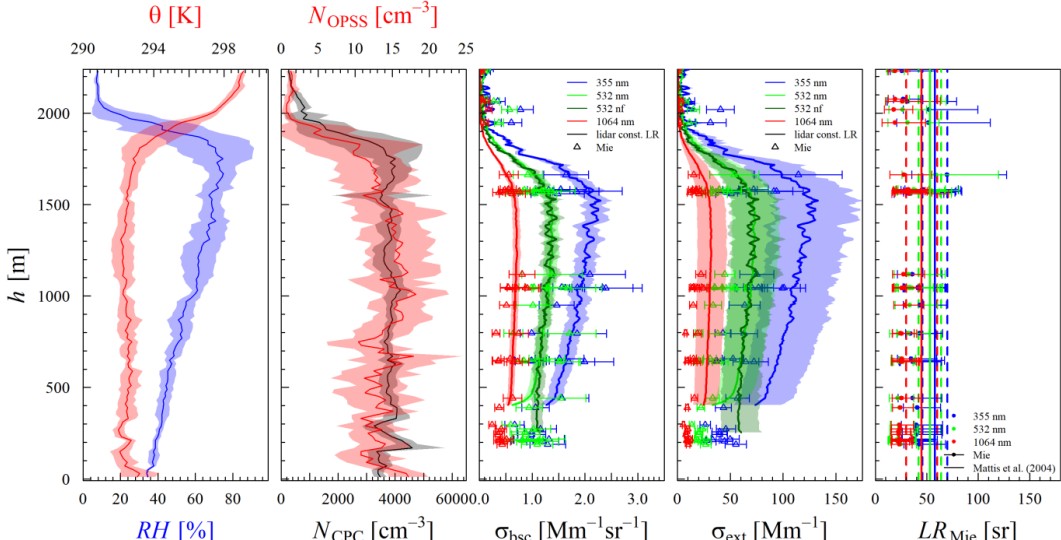

**Figure 8: Same as Figure 6 for flight b on June 17, 2015, between 12:43 and 14:19 UTC.**

Figure 8 displays vertical profiles of the same observed parameters as shown in Figure 6 obtained during the second
flight (12:43 to 14:19 UTC) on June 17, 2015. Differently to June 26, a larger decrease of *RH* was observed above the top of
the PBL at around 1800 m to 2000 m altitude. Below 2000 m altitude, the *RH* was steadily decreasing from 75% to 35%
towards ground. The stable $N_{OPSS}$ and $N_{CPC}$ of ~15 cm⁻³ and 3800 cm⁻³, respectively, indicated a well-mixed planetary boundary
layer up to an altitude of around 1800 m. Opposing to the case of June 26, 2015, on average, the modeled values of the
$\sigma_{bsc}(\lambda)$ were lower than the lidar-based ones by 1.4% to 12.3% (see Table 2). The model calculated significantly lower (42.9%
to 35.9%) $\sigma_{ext}(\lambda)$ in the ambient state than derived with the lidar using the $LR(\lambda)$ of Mattis et al. (2004).

We assume that the *LR*s for urban haze aerosol reported by Mattis et al. (2004) might not apply to that day. The
spectral behavior of $LR_{Mie}(\lambda)$ was different from the case of June 26. In particular, during flight b on June 17, the $LR_{Mie}$(532 nm)
was in the range of $LR_{Mie}$(1064 nm), whereas on June 26 $LR_{Mie}$(532 nm) was in the range of $LR_{Mie}$(355 nm). Within the
lowermost 400 m, under dry conditions at around 40% *RH*, the $LR_{Mie}$(355 nm) was around 38 sr, at $LR_{Mie}$(532 nm) and
$LR_{Mie}$(1064 nm) was around 23 sr. This is agrees with Catrall et al. (2005) who have reported a *LR*(550 nm) of 28 (±5) sr with
a ratio of *LR*(550 nm)/*LR*(1020 nm) of 1.0(±0.2) for marine aerosol. Hence, the prevalent aerosol on this day possibly could
be classified as marine type aerosol applying the classification of Catrall et al. (2005). The origin of the corresponding
trajectory cluster (WS-A2 (clean); Sun et al., 2019) located over the North Atlantic support this aerosol classification. Applying
the $LR_{Mie}(\lambda)$ displayed in the fifth panel to $\sigma_{bsc,lid}(\lambda)$, the slope of the linear fit of modeled and the lidar-based $\sigma_{ext}(\lambda)$ was much
closer to 1 and the agreement was within 12.9% (underestimation of 7% at 1064 nm, 7.9% at 532 nm, 5.2% at 532 nm near-
field channel, and 12.9% at 355 nm).

Averaged over all four investigated flights, the Mie-model calculated lower optical coefficients than derived by the
lidar. Table 2 summarizes the slopes of the correlation between measured and modeled optical coefficients of the four
investigated flights.






**Table 2: Overview of the slopes and their standard error of a linear regression between the modeled extinction and backscattering coefficient with the measured ones from the lidar for the four investigated flights and summarized for all data points display with three significant figures accuracy.**

| flight | backscattering | | | extinction | | |
|---|---|---|---|---|---|---|
| | 355 nm | 532 nm<br>532 nm NF | 1064 nm | 355 nm | 532 nm<br>532 nm NF | 1064 nm |
| 17b | 0.877 (±0.046) | 0.963 (±0.0568)<br>0.986 (±0.0552) | 0.932 (±0.0484) | 0.641 (±0.0386) | 0.578 (±0.0315)<br>0.588 (±0.0331) | 0.571 (±0.0295) |
| 26a | 0.647 (±0.0333) | 0.809 (±0.0401)<br>0.858 (±0.0455) | 1.34 (±0.064) | 0.833 (±0.0316) | 1.05 (±0.0416)<br>1.12 (±0.049) | 1.31 (±0.0583) |
| 28a | 0.706 (±0.0295) | 0.709 (±0.0363)<br>0.588 (±0.0352) | 0.577 (±0.035) | 0.562 (±0.0293) | 0.568 (±0.0383)<br>0.482 (±0.03) | 0.411 (±0.031) |
| 28b | 0.583 (±0.0369) | 0.774 (±0.045)<br>0.834 (±0.059) | 0.638 (±0.0379) | 0.495 (±0.0504) | 0.566 (±0.0486)<br>0.627 (±0.0509) | 0.463 (±0.0316) |
| all | 0.678 (±0.019) | 0.825 (±0.0226)<br>0.837 (±0.0258) | 0.908 (±0.0363) | 0.748 (±0.0205) | 0.864 (±0.0292)<br>0.871 (±0.0336) | 0.711 (±0.0388) |

610         On average, the modeled $\sigma_{bsc}(\lambda)$ was 32.2 (±1.9)% lower at 355 nm, 17.5 (±2.3)% at 532 nm, 16.3 (±2.6)% at 532 nm near-field channel, and 9.2 (±3.6)% lower at 1064 nm; the modeled $\sigma_{ext}(\lambda)$ was 25.2 (±2.1)% lower at 355 nm, 13.6 (±2.9)% at 532 nm, 12.9 (±3.4)% at 532 nm near-field channel, and 28.9 (±3.9)% lower at 1064 nm. Ferrero et al. (2019) have shown that unaccounted dust has a significant impact on the modeling of $\sigma_{bsc}(\lambda)$. Their Mie-calculations have been 72% to 39% lower than the corresponding lidar measurements without considering dust. After considering the 45% of unaccounted $PM_{10}$ mass as

dust, the modeled results agreed with the lidar measurements (37% overestimation at 355 nm, and within 7% at 532 nm and 1064 nm) and increased the intensity of the scattered light at 180° significantly. In this study we do not consider dust or any other crustal material within the chemical composition. Hence, the missing dust and crustal material could explain the underestimation of the Mie-model.

        Another reason could be an underestimation of the aerosol hygroscopicity and hence an underestimation of the aerosol

particle growth resulting in a lower simulated extinction and backscatter cross-section of the aerosol particles in ambient state. As stated by Wu et al. (2013) evaporation of $NH_4NO_3$ within the VH-TDMA system can occur and therefore the hygroscopicity is underestimated compared to size-segregated hygroscopicity estimates based on chemical composition measurements. Also, as shown by Rosati et al (2016b), the variation in temperature and *RH* can have an influence on the apportionment of ammonium nitrate which has a $\kappa$ of 0.68 (see Appendixtable 1). A lower temperature at higher altitudes results in less

evaporation and thereby to a larger volume fraction of ammonium nitrate and hence to a larger hygroscopicity in that altitude.

        Furthermore, De Leeuw and Lamberts (1986) have showed that $\sigma_{bsc}(\lambda)$ is sensitive to a) the refractive index and b) covered size-range. At a size-constant imaginary part of 0.05 the variation in $\sigma_{bsc}(\lambda)$ for a real part of 1.4 to 1.6 is almost one order of magnitude. At a real part of 1.56, they have shown that increasing the imaginary part from $10^{-3}$ to $10^{-1}$ decreases $\sigma_{bsc}(\lambda)$ by one to two orders of magnitude. Since the imaginary part is mainly driven by the BC content within the aerosol, an

overestimation of the BC mass would result into a larger imaginary part of the refractive index and hence to a $\sigma_{bsc}(\lambda)$ which would be too small. Also, they stated, extending the covered aerosol particle diameters to more than 32 µm significantly increases both extinction as well as backscatter. They also showed that $\sigma_{ext}(\lambda)$ is in general less sensitive to the imaginary part


complex refractive index compared to $\sigma_{bsc}(\lambda)$. However, the real part is important and the aerosol particle light extinction increases with increasing real part. Thereby, the increase is larger the smaller the wavelength is. Hence, a) non-captured aerosol

particles larger than the observed size-range could led to larger $\sigma_{bsc}(\lambda)$ and $\sigma_{ext}(\lambda)$, and b) the constant complex aerosol refractive index over all wavelengths and for all particle sizes could also had an influence on the results. However, the bulk chemical composition approach has shown good agreements with the in-situ scattering measurements on ground – at least at 450 nm. A wavelength-dependent complex refractive index of the aerosol components could improve the agreement.

Furthermore, the approach of correcting the airborne PNSD with the OPSS-based altitude correction factor $f_h$ might

underestimates $dN/dlogD_p$ in higher altitudes which would result into lower modeled optical coefficients than observed with the lidar.

Ma et al. (2012) has already shown, that a mixture of fully externally and internally core-shell mixed aerosol containing light absorbing carbon is a better representation to derive the hemispheric aerosol particle light backscattering coefficients (HBF) and they reported a mass fraction of fully externally mixed light absorbing carbon of 0.51 (±0.21) for in

the North China Plain for July 12 to August 14, 2009. With fixed refractive indices of the aerosol components (1.8 + 0.54i for light absorbing carbon and for the less absorbing components 1.55 + 1e-7i) and constant volume fractions for the whole observed particle size range, they have shown that the core-shell approach overestimates the measured HBF at 450 nm by around 10% and underestimates the measured HBF by about 5% at 700 nm wavelength. Although HBF is not $\sigma_{bsc}(\lambda)$, these results show that the constant mixing approach in this study might led to biases in the modeled aerosol optical coefficients.

**4.1.2 *RH* dependence of the *LR*($\lambda$)**

Based on the four measurement flights during the summer campaign, the $LR(\lambda)$ dependence on the $RH$ have been examined. The winter cases have been excluded in this analysis because the underlying measurements were, although basically based on airborne in-situ measurements, different in a) the underlying hygroscopicity estimates, and b) the measured aerosol particle number size distribution.

The fifth panel of Figure 6 and Figure 8 displays the Mie-based ambient state $LR(\lambda)$ at the given wavelengths (dots with error bars) and the reference $LR(\lambda)$ of Mattis et al. (2004), represented by the color-coded vertical lines with the given uncertainty range marked as dashed lines around these. The mean $LR(\lambda)$ of flight 26a calculated with the Mie-model in the ambient state was 64.1 (±14.1) sr at 355 nm, 61.7 (±10.9) sr, and 36.2 (±8.0) sr at 1064 nm which is 10.5% larger, 16.4% larger and 19.6% lower than the corresponding $LR(\lambda)$ reported by Mattis et al. (2004) but in the given range. The vertical structure

of $LR_{Mie}(\lambda)$ did follow the trend of the $RH$.

Previous studies reported a significant influence of the $RH$ on the aerosol optical properties often expressed with an enhancement factor. Zieger et al. (2013), e.g., presented the aerosol particle light scattering enhancement for different European sites, Skupin et al. (2016) published a four-year-long study on the impact of the $RH$ on the aerosol particle light extinction for Central European aerosol, and Haarig et al. (2017) showed the backscatter and extinction enhancement for marine aerosol.

Ackermann (1998) investigated the dependence of the $LR(\lambda)$ on $RH$ for different aerosol types with a numerical simulation, but has not presented a $LR(\lambda)$ enhancement factor and the underlying PNSD were solely based climatology data and not based on actual measurements like within this study. Following the approach of Hänel (1980) the $RH$- and wavelength-dependent enhancement factor of the $LR(\lambda)$, $f_{LR}(RH, \lambda)$, is expressed with:

$$f_{LR}(RH, \lambda) = f_{LR,\text{dry}} \times (1 - RH)^{-\gamma(\lambda)}, \tag{9}$$

where $f_{LR,\text{dry}}$ is equal to $f_{LR}(RH = 0, \lambda)$, the $LR(\lambda)$ enhancement factor at 0% $RH$ and is forced through 1. $\gamma(\lambda)$ denotes the wavelength dependent fitting exponent.



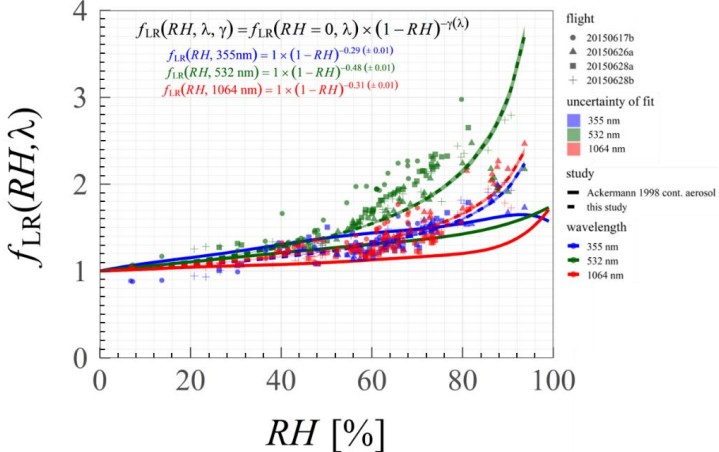

**Figure 9: Mie-based *RH*-dependent *LR*(λ) enhancement factor $f_{LR}(RH, λ)$ calculated with the airborne in-situ PNSD derived with ACTOS plotted for the three lidar wavelengths (dashed line). Symbols represents the investigated flights, colors the wavelength, and the shaded area around the standard error of the fit. In comparison, the estimates for continental aerosol of Ackermann (1998) translated into the lidar ratio enhancement factor is displayed as solid lines.**

The estimated $f_{LR}(RH, λ)$ for the four investigated measurement flights (17b, 26a, 28a, 28b) is displayed in Figure 9 and Table 3 shows the corresponding fitting parameters with the standard errors of the fit. Note that the "dried state" *LR*(λ) was calculated for aerosol with some residue water, because the sampled aerosol was never completely dry. The *RH* measured after the dryer was at most 48.3% on flight 20150617b and reached a maximum of 35.8% on the other days. In the Mie-model the aerosol particles in dried state were treated as completely dry. However, the growth in size of the aerosol particles at this *RH* level is small (around 10%) and the bias on the *LR*(λ) enhancement estimates should be negligible small.

The *LR*(λ) enhancement factor shows a clear dependence on the ambient *RH* with an expected enhancement factor of around one at low *RH*. The observed trend follows the results reported by Ackermann (1998) (solid lines in Figure 9) for continental aerosol but with larger quantities especially at larger *RH*. Also, the aerosol sampled in this study resulted in a *LR*(λ) enhancement factor of up to 3.7 at 532 nm and up to 2.4 (2.2) at 1064 nm (355 nm) at 93.7% *RH*. The power series representation of Ackermann (1998) resulted in a $f_{LR}$(355 nm) of 1.6, $f_{LR}$(532 nm) of 1.73, and $f_{LR}$(1064 nm) of 1.71 at 99% *RH*.

$f_{LR}(RH, 355$ nm$)$ and $f_{LR}(RH, 1064$ nm$)$ behave similar. The calculated *LR* enhancements of each day follow the overall trend but the data points of flight 20150617b, indicated filled circles, have shown a positive offset to the fitting function. A predominant aerosol type at that day, which was different to the other shown days, is assumed to be the reason of a different *LR*(λ) enhancement factor behavior.

$γ(532$ nm$)$ is significant larger than $γ(355$ nm$)$ and $γ(1064$ nm$)$, respectively. The data-points sampled under ambient conditions of 60% to 80% *RH* are overrepresented in the fit. Furthermore, Mie calculations (settings: $f_{v,eBC} = 0.03$, $κ = 0.3$, T = 20°C, core-shell mixture), conducted on the basis of the PNSD measured at Melpitz Observatory during June 26, 2015, have shown that in this *RH* range the *LR*(532 nm) gets more enhanced than the *LR*(1064 nm) or *LR*(355 nm) and might be a typical feature of the predominant aerosol or results from the model constraints. Similarly, in the results of Ackermann (1998) the *LR*-to-*RH* dependence for continental aerosol was not following the exponential curve perfectly. Also, *LR*(λ) for marine aerosol is more enhanced at this *RH* range as reported by Ackermann (1998). The fit for 532 nm at this *RH* range, therefore, might was over-weighted which might led to an overestimation of $γ(532$ nm$)$. Also, at 355 nm Ackermann (1998) has shown





a decreasing $LR$(355 nm) above 90% $RH$ which we could not observe in this study because of a small number of cases and the observed $RH$ range.

The results are opposed to the findings of Takamura and Sasano (1987), showing a negative correlation of $LR(\lambda)$ and $RH$ at 355 nm and a small dependence of the $LR(\lambda)$ on the $RH$ at larger wavelengths. This might be caused by their different analysis approach since Takamura and Sasano (1987) used PNSDs inferred from angular light scattering measurements of a polar Nephelometer including more uncertainty-increasing processing steps. Also, their Mie calculations were based on PNSD estimates at different $RH$ levels with assumed homogeneously mixed aerosol particles with an effective complex refractive index at ambient state. Contrary, our investigations based on hygroscopic growth simulations and a core-shell mixing approach. Furthermore, the limited covered size-range of the aerosol particle hygroscopicity might introduces some bias in our results since the $\kappa(D_{\mathrm{p}})$ estimates above 265 nm are maybe too large or too small, which would have an impact on the Mie-model results, especially on $\sigma_{\mathrm{bsc}}$, which is more sensitive to the complex aerosol refractive index than $\sigma_{\mathrm{ext}}(\lambda)$.

Nevertheless, the presented results provide good first estimates of the $RH$-induced $LR(\lambda)$ enhancement factor based on in-situ measured PNSD for the observed $RH$ range. Although Ackermann (1998), already, has shown the $LR$-to-$RH$ dependence for three different aerosol types (marine, continental, dessert dust), future research should collect more data to provide $f_{\mathrm{LR}}(RH, \lambda)$ with the corresponding $\gamma(\lambda)$ estimates including a separation into different aerosol types.

Future research should investigate the impact of the mixing-state and hygroscopic growth factor representation within the Mie-model on the lidar ratio enhancement factor as well.

**Table 3: Overview of the fitting parameter of the $LR(\lambda)$ enhancement factor. The standard error of fit is marked with brackets.**

| wavelength $\lambda$ [nm] | $\gamma(\lambda)$ |
|---|---|
| 355 | 0.29 (±0.01) |
| 532 | 0.48 (±0.01) |
| 1064 | 0.31 (±0.01) |





### 4.2 MelCol-winter

Data representing another season with different atmospheric conditions was collected and evaluated for the winter of 2017. Exemplarily, the data of two measurement days within winter 2017 is discussed in the following.


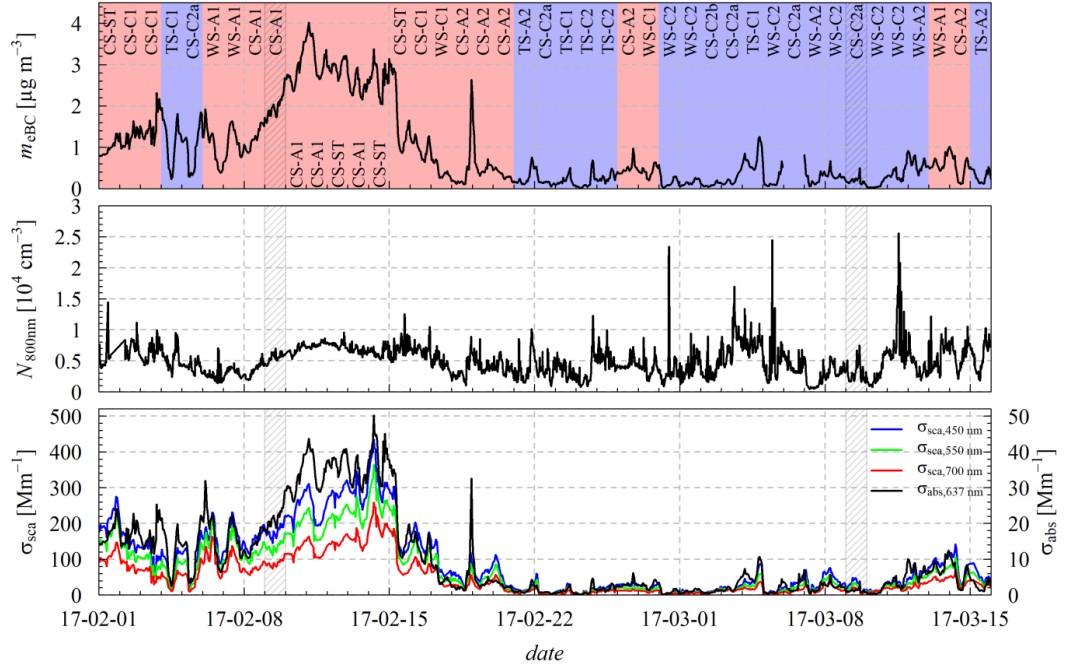

**Figure 10: Corresponding to Figure 5 for the period February 1 to March 15, 2017.**

Figure 10 shows the period from February 1 to March 16, 2017, which was characterized by two distinct periods.

Period 1 from February 1 to about February 19 was featured by a high $m_{eBC}$ of up to 4.0 µg m$^{-3}$, while the subsequent period was relatively clean. This maximum $m_{eBC}$ was in the same range than the observed maximum rBC mass concentration ($m_{rBC}$) of 4 µg m$^{-3}$ reported by Yuan et al (2020) at Melpitz during the period between February 1 and February 19, 2017. Long-term measurements at Melpitz in the period from 2009-2014 reported by Birmili et al (2016) were characterized by an average $m_{eBC}$ of 0.9 µg m$^{-3}$. Trajectory cluster CS-A1 and CS-ST, both categorized as polluted, are assigned to the period from February 8

to February 15. In this period, the air masses were transported from northern Ukraine crossing southern Poland (cluster CS-A1; Sun et al., 2020), a hotspot of elemental carbon emissions (see 7x7 km-EUCAARI EC-emissions in Chen et al., 2016). The combination of airmass origins (East Europe and stationary) resulted in an accumulation of pollution over Melpitz. The mass concentration of aerosol particles with an aerodynamic diameter lower than 2.5 µm (PM$_{2.5}$) on February 9 exceeded typical annual average PM$_{2.5}$ aerosol particle mass concentrations (e.g. Spindler et al, 2013; 20.1 ± 18 µg m$^{-3}$) by a factor of

two and illustrates the unusually high pollution during this period. The measurement days February 9 and March 9, 2017, investigated in this paper, are highlighted with the gray shaded area in all three panels. Both days represent different atmospheric conditions and are discussed in more detail.



### 4.2.1 Optical closure of Mie-model and lidar during MelCol-winter

**Aerosol Particle Light Absorption**

During winter, two balloon launches during different levels of pollutions were conducted. This part focuses on the evaluation of the model with airborne in-situ measurements in a dried state. The corresponding atmospheric conditions are shown. The findings provide insights to, e.g., evaluate $\sigma_{abs}(\lambda)$ derived from lidar with similar setups.

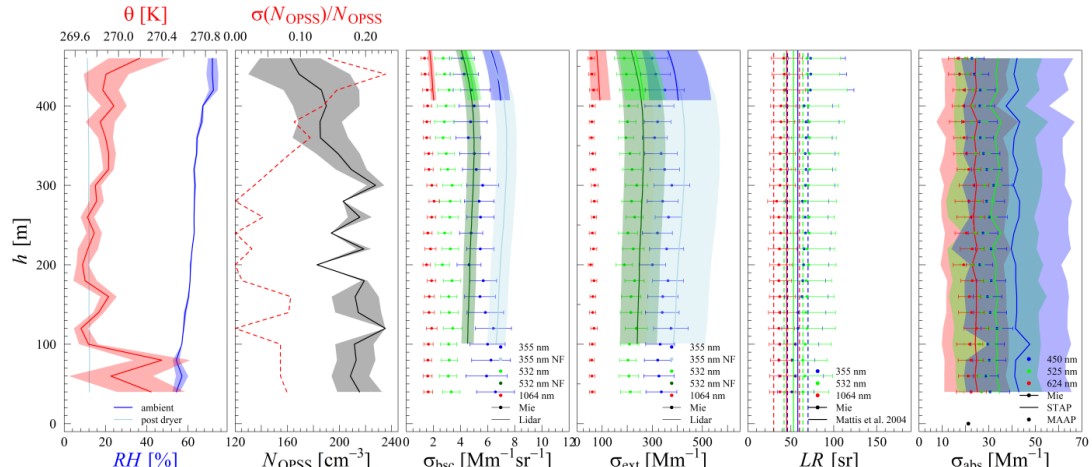

**Figure 11: 20-m layer averages of the ambient, and post dryer *RH* and *T* (first panel), the aerosol particle number concentration measured by the OPSS (*N*OPSS), and the ratio of the standard deviation of mean and the mean (solid black and red dashed line, second panel). Shaded areas around *T*, *RH*, and *N*OPSS represent the standard deviation of the mean in the layer. Also, the aerosol particle light backscattering ($\sigma_{bsc}(\lambda)$, third panel), extinction ($\sigma_{ext}(\lambda)$, forth panel), and absorption coefficients ($\sigma_{abs}(\lambda)$, sixth panel) are shown. Mean values are calculated for the period 11:20-11:58 UTC on February 9, 2017. Shaded areas in the sixth panel**
**represent the standard deviation of mean. Shaded areas around the lidar-based coefficients indicate the assumed 10% uncertainty of $\sigma_{bsc}(\lambda)$ and the range of possible $\sigma_{ext}(\lambda)$ following the given range of Mattis et al. (2004). The fifth panel displays the *LR*($\lambda$) derived with the Mie model (dots with a range bar from min to max) and the reference of Mattis et al. (2004) with its respective uncertainty range displayed with dashed lines. Uncertainty bars around the Mie-based coefficients cover the range from minus three to plus three time standard deviation Uncertainty around the *LR*($\lambda$) is minimum and maximum *LR*($\lambda$) resulting from calculations with the**
**threefold standard deviation from the $\sigma_{bsc}(\lambda)$ and $\sigma_{ext}(\lambda)$..**

Figure 11 displays the vertical distribution of 20-m averages of the ambient *RH* (blue line), post-dryer *RH* (light blue line), and *T* (red line) measured on February 9, 2017, between 11:20 and 11:58 UTC (first panel), the same time window of the averaged lidar profiles. This measurement day was characterized by a very sharp inversion which the balloon was not
capable to ascent through. Below, the atmosphere was in a well-mixed state indicated by a rather constant potential temperature of around 270 K and a stable *N*OPSS (second panel). NOPSS was varying in the range of 180 cm⁻³ to 220 cm⁻³ within the lowermost 300 m above ground followed up by a steady decrease to around 160 cm towards 450 m altitude. Panel three and four display the modeled and lidar-based $\sigma_{bsc}(\lambda)$ and $\sigma_{ext}(\lambda)$.

Figure 12 displays the vertically resolved atmospheric parameters also shown in Figure 11 for March 9, 2020, between
13:30 and 14:09 UTC. Compared to February 9, March 9 was characterized by a much lower atmospheric aerosol load within the PBL indicated by an almost three times lower N$_{OPSS}$. The measurement flight during this day could profile the atmosphere up to an altitude of around 1080 m and hence the entire planetary boundary layer was covered. The top of the PBL was reaching an altitude of around 750 m indicated by the temperature inversion at this height (see Figure 12 first panel).

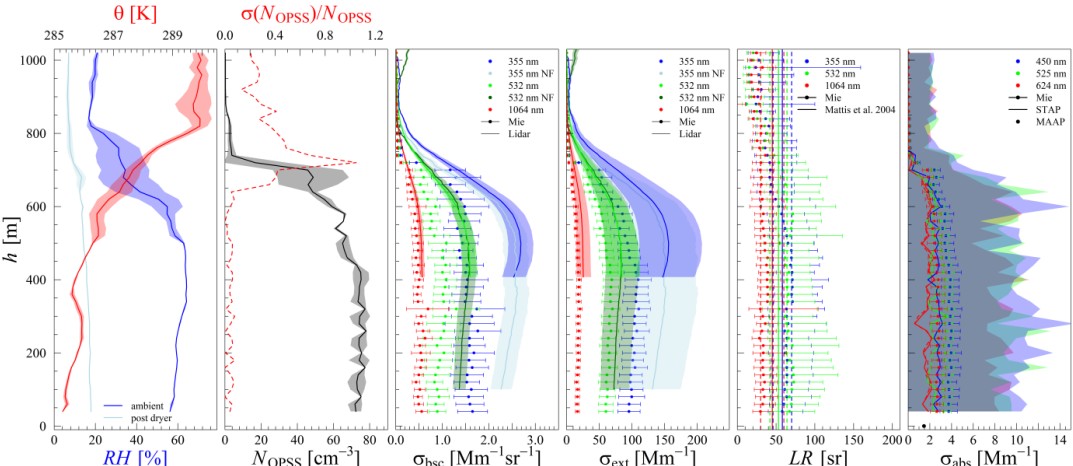

**Figure 12: Corresponding to Figure 11 for the period 13:30-14:09 UTC on March 9.**

The profiles of the Mie-modeled and measured $\sigma_{abs}(\lambda)$ in dried state conducted on February 9 and March 9, 2017, are shown in the last panel of Figure 11 and Figure 12. The linear fit and the corresponding fittings are displayed in Figure 13, Figure 14, fitting parameters are given in Table 4.

On February 9 between 11:00 and 12:00 UTC and March 9 between 13:00 and 15:00 UTC, the MAAP on ground measured a mean $\sigma_{abs}(637\ nm)$ of 21.2 Mm$^{-1}$ and 1.46 Mm$^{-1}$, respectively (Figure 11 and Figure 12; black dot panel 6) which was 12.9% and 15.2% lower than the average $\sigma_{abs}(624\ nm)$ measured by the STAP within the lowermost 200 m above ground (24.3 Mm$^{-1}$, 1.7 Mm$^{-1}$).

The spectral behavior of the $\sigma_{abs}(\lambda)$ can be described with the absorption Ångström exponent *AAE*:

$$AAE(\lambda_1,\lambda_2) = -\frac{\ln\left(\frac{\sigma_{abs}(\lambda_1)}{\sigma_{abs}(\lambda_2)}\right)}{\ln\left(\frac{\lambda_1}{\lambda_2}\right)}. \tag{10}$$

The $AAE_{STAP}(624\ nm, 450\ nm)$ was $1.67 \pm 0.14$ on average within the lowermost 700 m on February 9, and was slightly larger than the daily mean $AAE_{AE33}(660,450\ nm)$ of 1.49 (±0.08 standard deviation of mean) derived from parallel conducted, spectrally resolved, $\sigma_{abs}(\lambda)$ measurements of an Aethalometer at Melpitz (model AE33; Magee Scientific, Magee Scientific, Berkeley, CA, USA). For March 9, 2017, we could not compare the *AAE* since the AE33 was stopping its measurements on February 22, 2017. The comparison of the $AAE_{STAP}(624\ nm, 450\ nm)$ with $AAE_{AE33}(660,450\ nm)$ and of $\sigma_{abs,STAP}(624\ nm)$ with the MAAP indicated a good representation of the $\sigma_{abs}(\lambda)$ derived by the STAP. The comparison of the measurements of the MAAP and AE33 in the period between February 4 and February 22, 2017, revealed a dependence of $\sigma_{abs,AE33}(635\ nm) = 1.27\ \sigma_{abs,MAAP}(637\ nm)$.

As shown in Figure 3b), in the winter period, the Mie-model simulated on average around 8% larger $\sigma_{abs}(637\ nm)$ than measured by the MAAP. For the airborne measurements, the assumptions within the Mie-model to derive $\sigma_{abs}(\lambda)$ in the dried state led to a 31.8 (±1.5%), 24.7 (±1.7%) and 13.2 (±1.7%) underestimation at 450 nm, 525 nm, and 624 nm respectively on February 9. On March 9, 2017, a 32-37% overestimation of the airborne measured $\sigma_{abs}(\lambda)$ was observed (see Figure 13, Figure 14; corresponding profiles in Figure 11 and Figure 12). This indicates a spectral dependence.

At ground, the Mie-simulation based on the aerosol microphysical measurements calculated a $\sigma_{abs,Mie}(630\ nm)$ on February 9 (March 9) which was 12.8% (103%) larger than measured by the MAAP at 637 nm. The assumptions within the model who led to the overestimation of the ground-based $\sigma_{abs}(\lambda)$ estimates propagated into the airborne modeling. An overestimation of 103% indicates aerosol conditions during March 9 which could not be captured by the model. For instance, the estimated *MAC*(637 nm), which indirectly leads to the eBC volume fraction used within the model, could have been too





small as a result of probably too small $m_{EC}$ measurements. However, we considered EC as eBC, which could have led to some
bias in the $MAC$(637 nm) estimate as well. In particular, on February 9, a $MAC$(637 nm) of 10.9 m$^2$ g$^{-1}$ was derived, on March
9, a small $MAC$(637 nm) of 6.6 m$^2$ g$^{-1}$. The time-series of the $MAC$(637 nm) estimates are displayed in Appendixfigure 1.

Zanatta et al. (2018) and Yuan et al (2020), e.g., have shown that the mixing of BC is an important parameter
influencing directly the value of the $MAC(\lambda)$. They reported $MAC(\lambda)$ for pure externally mixed BC aerosol particles. For
Melpitz, during the winter period of this study and applying an $AAE$ of 1, the $MAC$(870 nm) of 5.8 m$^2$ g$^{-1}$ reported by Yuan et
al. (2020) translates into 7.9 m$^2$g$^{-1}$ at 637 nm. With an $AAE$ of 1, modeled $MAC$(550 nm) for pure BC particles reported by
Zanatta et al. (2018). translate into very small 3.5 m$^2$ g$^{-1}$ to 5.7 m$^2$ g$^{-1}$ at 637 nm depending on the particle size. Nevertheless,
the $MAC$(637 nm) on February 9, coincided with the estimates of Yuan et al. (2020). Therefore, on February 9, 2017,
$\sigma_{abs,Mie}$(624 nm) and $\sigma_{abs,STAP}$(624 nm) agree reasonably well within 13.2% since a $MAC$ estimated at 637 nm represents 624 nm
reasonably well.

The core-shell mixing representation within the model was not applicable to the aerosol on March 9, because a
$MAC$(637 nm) in the range of the estimates of Yuan et al. (2020) and Zanatta et al. (2018) indicates external mixture rather
than an internal core-shell mixture. The larger $MAC$(637 nm) on February 9, on the other, hand suggest a good representation
of the mixing state of the prevalent aerosol.

The spectral dependence of the over- and underestimation for both days can be explained with the $AAE$. Within the
lowermost 700 m above ground, a median $AAE_{Mie}$(624 nm, 450 nm) of 0.94 was found; on February 9, and of 1.05 on March
9, respectively. The corresponding median $AAE_{STAP}$(624 nm, 450 nm) of 1.64 on February 9, and of 1.20 on March 9, clearly
indicated a significant amount of BrC aerosol particles according to Zhang et al (2020). The $AAE$ of BC is near unity at visible
and near-infrared wavelengths (e.g., Kirchstetter and Thatcher, 2012) but also can go as high as 1.6 when BC is coated with
transparent material as stated by Cappa and Lack (2010). The values of $AAE_{Mie}$(624 m, 450 nm) of around 1 agree with these
findings. $AAE_{STAP}$ on both days, and $AAE_{AE33}$ on February 9 indicated the presence of BrC. BrC contributes less to the
absorption at near-infrared wavelengths with increasing contribution to the aerosol particle light absorption towards UV
wavelengths (e.g. Kim et al., 2020 and Sun et al., 2007). The daily mean volume fraction of organic material detected by the
Q-ACSM on February 9 was 45.1% peaking at around 50% during the flight time. On March 9 during flight time, a volume
fraction of 34.4% was found with values as small as 17% in the morning hours. The small volume fraction (March 9) had less
of an impact on the Mie-model and led to the smaller spectral dependence of the overestimation. The larger volume fraction
on February 9, on the other hand, indicated a large content of BrC and hence a larger spectral dependence of the deviation.

To summarize, for March 9, it is more likely that a combination of the aerosol mixing representation within the model
as well as the possibly too small $MAC$(637 nm) led to the overestimation by the model rather than the missing BrC. For
February 9, the agreement within 13.2% at 624 nm indicated that the $MAC$(637 nm) represented the prevalent aerosol within
a satisfying range, the missing BrC content within the model, however, resulted into a larger spread in the underestimation.
The mixing approach within the model seemed to have better represented the aerosol present on February 9.

In conclusion, that future studies should a) consider the mixing state of the aerosol or at least include this in the
uncertainty analysis, and b) should include BrC with a spectral resolved $MAC(\lambda)$.

**Aerosol particle light backscattering and extinction coefficient**

Besides the in-depth view on the $\sigma_{abs}(\lambda)$, also a comparison of the lidar estimates of the $\sigma_{bsc}(\lambda)$ and $\sigma_{ext}(\lambda)$was conducted and
is shown below.


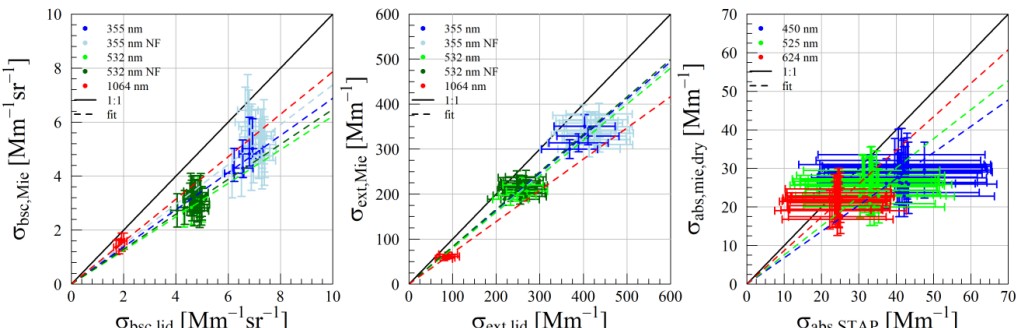

**Figure 13: Optical coefficients derived with the Mie model (ambient for extinction and backscattering; dry for absorption) based on the data from February 9 plotted against the coefficients derived with lidar and STAP, respectively. The black line indicates the 1:1 line and colors represent the respective wavelengths. Horizontal error bars indicate uncertainty range of the lidar estimates for backscattering, and extinction, for measured absorption they represent standard deviation of mean. Vertical error barsindicate three times the standard deviation of the mean in case of the Mie model.**

The $\sigma_{bsc}(\lambda)$ and $\sigma_{ext}(\lambda)$ are displayed in panels three and four of Figure 11 and Figure 12 for February 9, and March 9, 2017. The Mie-modeled coefficients are represented by dots with the three times standard deviation of the mean of the Mie-calculation, the lidar estimates as lines with the corresponding color.

Panel one and two of Figure 13 and Figure 14 display the correlation of the modeled and measured $\sigma_{bsc}(\lambda)$ and $\sigma_{ext}(\lambda)$ shown in Figure 11 and Figure 12 (panel three and four in each), correspondingly. The linear fit estimates, the corresponding standard error of fit and correlation coefficients are given in Table 4. Note that the shown fit of Figure 13 (Figure 14) is forced through the coordinate origin which artificially enhances the coefficient of determination $R^2$. The fits have been forced through zero since a) the range of the values of the observed optical coefficients was small and b) because both model and measurements rely on the present aerosol and if no aerosol is prevalent both, model and observation, should be zero. Therefore, results of $R^2$ should be considered with care.

For February 9, over all considered wavelengths, and field-of-view configurations of the lidar, the model results agreed within 21.2% to 37.8% (21.2% at 1064 nm to 37.8% at 523 nm; $R^2$ close to 1 in all cases) with the measured $\sigma_{bsc}(\lambda)$. The modeled $\sigma_{ext}(\lambda)$ were up to 30.5 (±1.8)% (at 1064 nm) lower than those derived based on the lidar measurements with a mean underestimation of 18.3 (±0.8)%. The approach of correcting the lower aerosol particles with the altitude correction factor might underestimated the aerosol particle number concentration of particles up to 300 nm. In Mie-theory, particles with about the same size of the incoming radiation wavelength are most efficient in scattering. In the study of Virkkula et al. (2011), aerosol particles in the range of 100-1000 nm contributed most to the aerosol particle light scattering at 550 nm. Therefore, at 355 nm an artificial under-sampling of the aerosol particles up to 300 nm in diameter induced by the altitude correction factor could have led to an underestimation in the modeled aerosol particle light scattering and thus extinction. Also, the Mie-model, as well as the correction of the OPSS, did not consider aspherical particles which could have led to a bias induced by the PNSD. Also, the wavelength-independent complex aerosol refractive index and probably, at this time present, non-captured, huge particles, as discussed already in the summer part, could explain some of the deviations. However, all modeled $\sigma_{ext}(\lambda)$ were within the range of the aerosol particle light extinction coefficients calculated with the minimum and maximum $LR(\lambda)$ provided by Mattis et al. (2004).

The fifth panel of Figure 11 shows the $LR(\lambda)$ with the range-bars indicating the minimum and maximum value of the result of the ambient state Mie modeling. Like in the summer cases, a clear connection between the increase of the $LR(\lambda)$ and the increase of the $RH$ was significant: with increasing $RH$ the $LR(\lambda)$ increased. Overall, the average $LR(\lambda)$ in the shown profile was 63.8 sr at 355 nm, 69.0 sr at 532 nm, and 37.6 sr at 1064 nm, which was in the range of the $LR(\lambda)$ reported by Mattis et al. (2004) except for the $LR(532)$ at 532 nm which was 7.8% larger than the maximum reported $LR(532)$. However, these





LR($\lambda$) seem reasonable since Catrall et al. (2005) reported a LR(550 nm) of around 70 sr for aerosol classified as urban/industrial aerosol and Omar et al. (2009) estimated a LR(532 nm) of 70 sr for aerosol classified as polluted continental
and smoke. Considering the origin of the aerosol (industrial area in south Poland) these results appear conclusive.

For March 9, 2017, the comparison of the Mie-calculations with the lidar-based estimates showed an underestimation at 1064 nm in backscattering of 14% ($0.86 \pm 0.02$) and in the case of extinction of 36% ($0.64 \pm 0.02$), respectively. In the case of backscattering, the underestimation increased with a decrease in wavelength and indicated that a wavelength-dependent complex refractive index is needed to precisely model $\sigma_{bsc}(\lambda)$. Overall the conditions have been relatively clean and were
similar to the shown cases of the summer campaign with roughly the same amount aerosol particle light absorption. The results of the summer have shown an underestimation of the lidar-estimates by the Mie-model with similar slopes of the linear fit as well. The assumption within the Mie-model in the dried state resulted in good agreement with in-situ measurements of $\sigma_{ext}(\lambda)$ and $\sigma_{sca}(\lambda)$, overestimating the in-situ measured $\sigma_{abs}(\lambda)$. However, the hygroscopic growth and the refractive index of the aerosol particles, estimated by their chemical composition, might have been inaccurate. Nevertheless, most of the modeled
$\sigma_{ext}(\lambda)$ matched with the lidar estimates within the range of the LR($\lambda$) estimates of Mattis et al. (2004). Except above 450 m altitude and 355 nm wavelength, where the modeled $\sigma_{ext}(\lambda)$ was significant smaller than the lidar estimates, which indicated an underestimation of the aerosol particle number concentration at this altitude and size-range due to an inaccurate altitude correction factor of the PNSD.

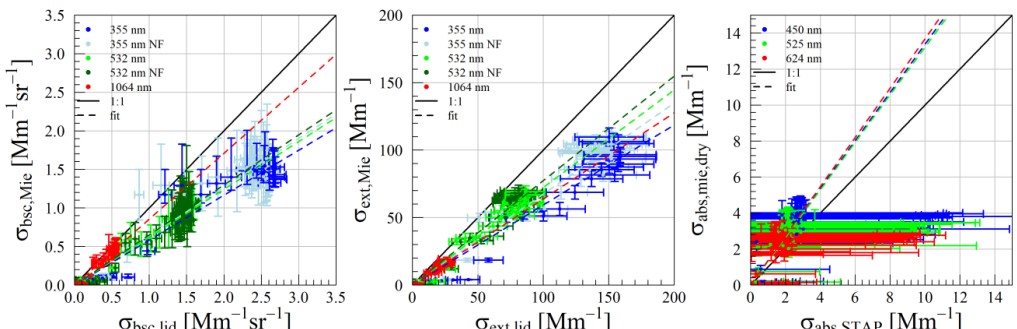

**Figure 14:  Corresponding to Figure 13 for the date of March 9, 2017.**


The calculated LR$_{Mie}$($\lambda$) is shown in the fifth panel of Figure 12. Within the planetary boundary layer, below 600 m altitude, where the ambient RH was stable, the LR$_{Mie}$($\lambda$) agreed within the estimates of Mattis et al (2004). At 355 nm a mean LR$_{Mie}$(355 nm) of 64.2 sr, at 532 nm a LR$_{Mie}$(532 nm) of 65.7 sr, and at 1064 nm a LR$_{Mie}$(1064 nm) of 34.3 sr was calculated indicating that the aerosol observed here was of type urban haze. Like in the profile of February 9, 2017, the vertical distribution
of the LR$_{Mie}$($\lambda$) did follow the trend of the ambient RH. The uncertainty of the LR$_{Mie}$($\lambda$) estimates increased with increasing standard deviation of the ambient RH as well.





**Table 4: Fitting estimates with its standard error and coefficients of determination ($R^2$) of the linear fits shown in Figure 13 and Figure 14. Abbreviation NF indicates the near-field channel of the lidar.**

| day | $\lambda$ [nm] | $\sigma_{bsc}$ | | $\sigma_{ext}$ | | $\sigma_{abs}$ | |
|---|---|---|---|---|---|---|---|
| | | $a$ | $R^2$ | $a$ | $R^2$ | $a$ | $R^2$ |
| 2017-02-09 | 355 | $0.69 \pm 0.02$ | 1.00 | $0.82 \pm 0.02$ | 1 | - | - |
| | 355 NF | $0.74 \pm 0.02$ | 0.99 | $0.81 \pm 0.01$ | 1 | - | - |
| | 532 | $0.62 \pm 0.01$ | 1.00 | $0.80 \pm 0.02$ | 1 | - | - |
| | 532 NF | $0.65 \pm 0.01$ | 0.99 | $0.83 \pm 0.01$ | 1 | - | - |
| | 1064 | $0.79 \pm 0.01$ | 1 | $0.70 \pm 0.02$ | 1 | - | - |
| | 450 | - | - | - | - | $0.68 \pm 0.01$ | 1 |
| | 525 | - | - | - | - | $0.75 \pm 0.02$ | 0.99 |
| | 624 | - | - | - | - | $0.87 \pm 0.02$ | 0.99 |
| 2017-03-09 | 355 | $0.58 \pm 0.02$ | 0.97 | $0.59 \pm 0.02$ | 0.98 | - | - |
| | 355 NF | $0.63 \pm 0.01$ | 0.98 | $0.67 \pm 0.01$ | 0.99 | - | - |
| | 532 | $0.62 \pm 0.01$ | 0.98 | $0.72 \pm 0.01$ | 0.99 | - | - |
| | 532 NF | $0.65 \pm 0.01$ | 0.98 | $0.77 \pm 0.01$ | 0.99 | - | - |
| | 1064 | $0.86 \pm 0.02$ | 0.98 | $0.64 \pm 0.02$ | 0.98 | - | - |
| | 450 | - | - | - | - | $1.34 \pm 0.06$ | 0.97 |
| | 525 | - | - | - | - | $1.32 \pm 0.04$ | 0.95 |
| | 624 | - | - | - | - | $1.37 \pm 0.06$ | 0.92 |


To summarize, the Mie-model reproduced a $\sigma_{ext}(\lambda)$ at ambient state closer to the lidar estimates at the more polluted case, whereas the in the clean case the underestimation was larger. In the case of $\sigma_{ext}(\lambda)$, no spectral trend was observed in terms of agreement indicating a bias induced by the PNSD rather than the by the complex aerosol refractive index. At 1064 nm, also, the Mie-model results were closest to the measured $\sigma_{bsc}(\lambda)$. That might be a hint, that the correction approach of utilizing

an altitude correction factor for the ground in-situ PNSD measurements was not able to reproduce the PNSD aloft of Melpitz, at least in the lower size-ranges. Equivalent to the summer cases, also the findings of De Leeuw and Lamberts (1986) and Ferrero et al. (2019) may provide some explanation for the observed results. However, both, modeling and lidar estimates, underlay uncertainties so that not only the modeled results could have been too small, also the lidar estimates could have been too large, especially in the extinction where the $LR(\lambda)$ is subject to a large uncertainty range. The underlaying reasons are

speculative and many parameters within the model can be varied. For $\sigma_{bsc}(\lambda)$ and $\sigma_{ext}(\lambda)$, we do not suspect that the missing BrC within the model would result into significant different results. However, considering the limitations of the measurements setup, e.g., the limited covered size-range and no vertical resolved chemical composition measurements, the results are promising.

**5 Summary and Conclusion**

This study presented the comparison of lidar estimates of $\sigma_{bsc}(\lambda)$ and $\sigma_{ext}(\lambda)$ with airborne in-situ measurement-based modeled ones and examines the effect of the $RH$ to the aerosol particle light extinction-to-backscatter ratio. Also, it evaluated modeled $\sigma_{abs}(\lambda)$ with airborne measured ones in a dried state to determine whether the presented model can be utilized to evaluate lidar-based aerosol particle light absorption estimates. For this purpose, the results of two field campaigns carried out near Melpitz conducted in the summer of 2015 and February/March, 2017, covering different states of aerosol load, were utilized. In the

two campaigns, two different airborne systems were deployed to carry out aerosol in-situ measurements complemented by a





set of state-of-the art ground-based in-situ instrumentation as well as by a polarization Raman-lidar system directly measuring the aerosol particle light backscattering coefficient at three wavelengths. In this study a height-constant $LR(\lambda)$ was utilized to derive aerosol particle light extinction profiles from aerosol particle light backscattering profiles derived by the lidar system.

The in-situ measurements were used to calculate aerosol optical properties using Mie-theory. A core-shell mixture of the aerosol particles was assumed. The chemical composition of the aerosol particles measured on the ground was set constant over all particle sizes and was assumed to be representative for all altitudes above ground. The model validation under dry conditions confirmed the underlying assumptions with modeled values matching the in-situ measurements within 18%. An additional module of the Mie-model calculated the aerosol optical properties in ambient state utilizing a hygroscopic growth simulation based on Kappa-Köhler theory. In both campaigns the airborne-based PNSD was extended with height-extrapolated

ground-based in-situ PNSD measurements.

Mie-model results and lidar measurements were compared with each other. In the summer case, the Mie-model calculated aerosol optical coefficients up to 32% lower than the lidar estimates, in the winter campaign they have been up to 42% lower. In both, the summer and winter campaign, a spectral dependence in the slope of the linear fit of the modeled and measured $\sigma_{bsc}(\lambda)$ was observed, whereas in $\sigma_{ext}(\lambda)$ not. This agrees with previous studies who have shown that $\sigma_{ext}(\lambda)$ (major

fraction is $\sigma_{sca}(\lambda)$) is less sensitive to the complex aerosol refractive index than $\sigma_{bsc}(\lambda)$ and is more driven by the PNSD. The results were promising, since the $\sigma_{bsc}(\lambda)$ especially requires a very precise determination of the aerosol state in terms of PNSD and chemical composition (refractive index and mixing state).

In the winter campaign, the Mie-model result was directly compared to the filter-based airborne in-situ $\sigma_{abs}(\lambda)$ measurements. In the more polluted case, the Mie-model derived up to 32% lower $\sigma_{abs}(\lambda)$ with the best agreement at 624 nm

wavelength and a showed a distinct spectral dependence of the agreement. In the cleaner case, the Mie-model calculated up to 37% larger $\sigma_{abs}(\lambda)$ with a small spectral dependence. The results indicated that the mixing-state of the aerosol, the wavelength-dependent complex refractive index of the aerosol compounds, as well as the BrC content, must be accurately represented by the model to match the measured $\sigma_{bsc}(\lambda)$ within a narrow uncertainty-range.

Utilizing a height-constant $LR(\lambda)$ is widely applied to determine $\sigma_{ext}(\lambda)$ from $\sigma_{bsc}(\lambda)$ and the modeled $LR(\lambda)$ shown here

are in the range of $LR(\lambda)$ estimates presented by previous studies for different aerosol types. In both campaigns, the Mie-model ambient state calculations, however, revealed a dependence of the $LR(\lambda)$ to the ambient $RH$ and resulted in a $RH$ and wavelength-dependent $LR(\lambda)$ enhancement factor $f_{LR}(RH, \lambda) = f_{LR}(RH = 0, \lambda) \times (1 - RH)^{-\gamma(\lambda)}$, with $f_{LR}(RH = 0, \lambda)$ forced through 1. Estimates of $\gamma(\lambda)$ were derived based on the summer campaign data-set.

In conclusion:

a) Conducting closure studies of optical aerosol properties requires a precise determination of the aerosol mixing state, its composition, the inclusion of BrC, and the application of a wavelength-dependent complex refractive index.

b) Airborne in-situ measurements of, e.g. the aerosol chemical composition including the BrC content, would provide improvements in such studies and would allow to validate lidar-based $\sigma_{abs}(\lambda)$.

c) A wide range of aerosol particle sizes was covered within this study. However, the modeled $\sigma_{bsc}(\lambda)$ was lower than

the measured one. A much further extension of the observed aerosol particle size-range beyond 10 µm would ensure that this parameter would not cause such an underestimation based on the finding of the De Leeuw and Lamberts (1987).

d) Knowing the connection between $RH$ and the $LR(\lambda)$, the $LR(\lambda)$ enhancement factor would be a useful tool to estimate the $LR(\lambda)$ at ambient state, when the dry state $LR(\lambda)$ is known. Also, it allows to calculate back the $LR(\lambda)$ in dry state,

when the $LR(\lambda)$ is directly measured in ambient state and a $RH$ profile is known, e.g. by radio soundings.

e) However, long-term measurements must be conducted to verify the $LR(\lambda)$ enhancement estimates for various aerosol-types as well as different seasons.





**Appendix**

**Appendixtable 1: Density $\rho$ and hygroscopicity parameter $\kappa$ of the aerosol compounds to derive the volume fraction of each compound. Densities following [a]Lin et al. (2013) and references therein (Tang, 1996; Chazette and Louisse, 2001; Sloane, 1986; Haynes, 2011; Seinfeld and Pandis, 2006; Eichler et al., 2008), [b]Moteki et al. (2010), [c]Kreidenweis et al. (2008) and references therein (Tang and Munkelwitz, 1994; Marcolli et al., 2004), [d]Petters and Kreidenweis (2007), [e]Wu et al. (2013), [f]Zaveri et al (2010) and [g] Liu et al. (2014).**

| compound | density $\rho$ [g cm$^{-3}$] | $\kappa$ |
|---|---|---|
| $NH_4NO_3$ | 1.720[a] | 0.68[c] |
| $NH_4HSO_4$ | 1.780[a] | 0.56[c] |
| $(NH_4)_2SO_4$ | 1.760[a] | 0.53[d] |
| OM | 1.400[a] | 0.1[e,f] |
| BC | 1.800[b] | 0[e] |
| $NH_4Cl$ | 1.527[a] | 0.93[g] |
| $(NH_4)_3(SO_4)_2$ | 1.830[c] | 0.56[c] |


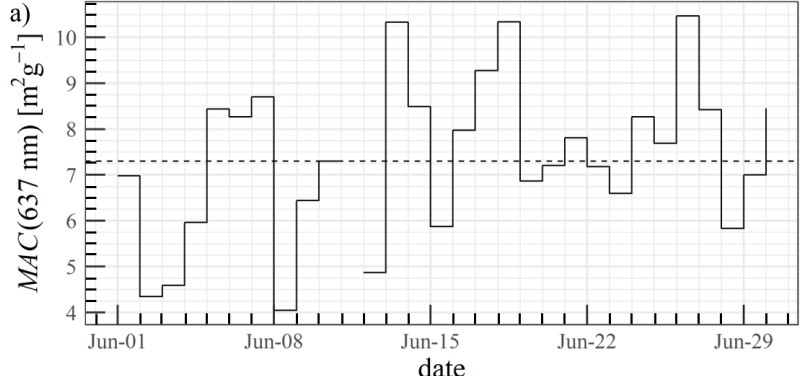

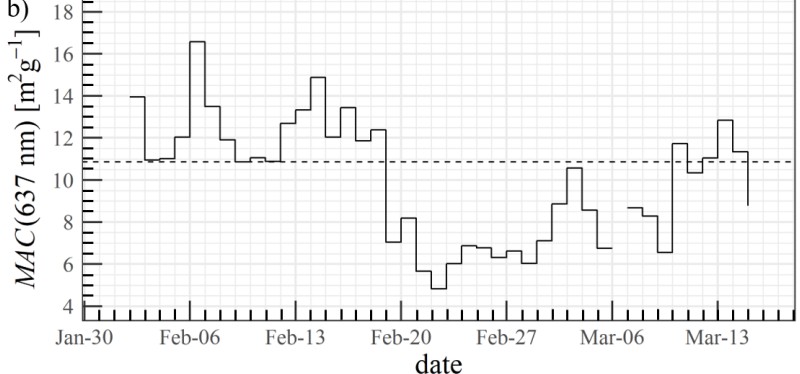

**Appendixfigure 1: $MAC$(637 nm) derived from measurements of the aerosol particle light absorption at 637 nm and mass concentration of elemental carbon at Melpitz Observatory. Horizontal dashed line indicates the median of the shown period. Panel**
**a) displays the period from June 01 to June 30, 2015. Panel b) displays February 1 to March 15, 2017.**



*Data availability.*

Data set and source codes underlying this work can be requested via email to the corresponding author.

*Authors contribution.*

The authors SD, BW, AA, and HB were responsible for the conceptualization of the study. Data curation, investigation, and the development of the methodology was done by SD. Further, for the study needed, data was provided by CD (V-HTDMA), GS (filter sampling data), LP (Q-ACSM), JCC (airborne CAPS data), TT (MPSS, APSS at Melpitz), TM (MAAP at Melpitz), and HB (lidar). Any software not included for processing was written by SD. The study was supervised by BW, TM, HB, BW, and AW. All figures were produced by SD. The original draft of the paper was written by SD. The review and editing of the
paper were done by SD, AA, HB, JCC, CD, MGB, TM, LP, GS, TT, BW, and AW.

*Competing interests.*

The authors declare that they have no conflict of interest.

*Acknowledgements.*

We gratefully thank the competent help of the technicians Thomas Conrath, Astrid Hofmann, and Ralf Käthner. We thank
Holger Siebert for setting up, and the built of ACTOS. We express our deepest thank to all other TROPOS employees supported us with energy and passion before, during, and after the campaigns and thank all participants helping to tame the balloon during the winter campaign. Moreover, we are very thankful to the helicopter pilots Alwin Vollmer and Jürgen Schütz for the secure helicopter flights during the summer campaign. The authors furthermore thank Dieter Schell of enviscope GmbH for his expertise. We also thank Anke Rödger of TROPOS for providing and conduction the filter-measurement samples of Melpitz.
JCC and MGB received financial support from the ERC (grant agreement no. 615922-BLACARAT) and from the ACTRIS2 project funded by the EU (H2020 grant agreement no. 654109) and the Swiss State Secretariat for Education, Research and Innovation (SERI; contract number 15.0159-1).





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
