# Peer review of "Measurement report: Comparison of airborne in-situ measured,"

_Atmospheric Chemistry and Physics, 2021_

## Referee Comment (RC2)

The article is still poorly written and tedious. Data from various measurements were used in this article including surface instrumentations like D-MPSS, Q-ACSM and lidar system. However, there are only lengthy analysis and few merits. Lidar has its advantage in measuring aerosol vertical hygroscopic growth in ambient environment, which is not evident in this article. In addition, errors due to the different observation methods and uncertainties from optical parameters from lidar should be considered.

Other comments:

The abstract is too long-winded and then innovation is difficult to find. Only one paragraph is suggested for the abstract.

Line 68: replace "does exsits" by "does exsit"

Line 101: replace "were" by "was"

Line 110: replace "is" by "are"

Line 112: replace "is" by "are"

Line 113: What is "This"?

Line 128: delete "is "

Line 230: replace "underestimates" by "underestimate"

Line 290: replace "of" by "for"

Line 343: replace "were" by "was"

Line 535: Poor writing

Line 578: replace "led" to "lead"

Line 618: replace "oppoding to" by "opposed to"

Line 702: "solely based on… "

---

## Author Comment (AC1)

Response to Referee #3:

*Blue italic font means the authors' answer,* standard, black *text symbolizes the reviewer's comments, and* **red italic font** *means manuscript changes, while* **black** *italic font symbolizes the original content of the manuscript.*

**Response:**

*We thank the referee for his fruitful comments, suggestions and questions. Below, we answered them point by point.*

Referee #3:

This study compares lidar measured optical properties to those computed with Mie calculations that used airborne in-situ based inputs over two different field campaigns. In general, the manuscript is unfocused and lengthy. The tedious amount of detail makes is difficult to completely comprehend and judge the merits of their analysis. I strongly suggest that the authors only include the details of the measurements that aren't described in other publications or are essential to the analysis performed in this work.

The authors state that their study is both "unique" and "complex", which it is, but why is it important? What science is advanced by this work? There are only a few lines of background/motivation given that mentions radiative forcing and cites the IPCC, but it is not clear how this work reduces uncertainty in radiative forcing.

Do the Mie-based calculations reproduce the optical properties well enough to meaningfully reduce radiative forcing uncertainty? The authors need give more a clear science motivation in the introduction and then revisit their goals in the conclusions to discuss what has been learned from this work.

*We removed the radiative forcing part of the motivation since it is not within the scope of the manuscript to improve radiative transport or radiative forcing estimates of aerosol. Instead, the study also aims to highlight the complexity of comparing multiple aerosol optical properties with in-situ and remote sensing techniques at once, particularly at the ambient state. Especially considering a high spatio-temporal resolution, such studies are often limited to payloads with a limited mass. Therefore, lightweight instruments are used preferably but with drawbacks in, e.g., the observed aerosol particle size range. Moreover, the determination of the composition of aerosol particles, which is necessary to determine their refractive index and hygroscopicity, is only possible with correspondingly expensive extensions of the load capacity of airborne systems. We updated the introduction and motivation towards the need for a comprehensive comparison study and what we can learn from the belonging challenges. Also, we think the title of the manuscript utilizing a closure study is misleading. We updated the title to "*
[revised manuscript text omitted]

As a means to reduce the scope/length of the manuscript, I would suggest that the authors consider removing the lidar parameterization analysis. For this analysis, it is hard to tell if the comparisons are a bit circular at times with in-situ inputs into a Mie model being used to derive the lidar ratio parameterization. That parameterization is then used derive the lidar extinction. Then the lidar extinction is compare to the in-situ measured extinction. It would make for the more straightforward comparison if the authors just made comparisons to the lidar backscatter

coefficients and avoided the assumptions/parameterizations needed to get the extinction altogether. This would also help shorten the manuscript. Plus, the lidar ratio parameterization is more a necessity because of the limitation of the Raman system to nighttime only and is less relevant to a general audience than the closure exercise.

*First of all, to address the amount of detail, we restructured the manuscript and shifted some parts to the supplementary material (Aerosol and atmospheric conditions during the campaigns, Mie model validation). Also, we wrapped the section of the used instrumentation merging the summer and winter campaign. Also, the scope of the manuscript is to provide additional information about the LR-RH-dependency and is, in our opinion, a valuable contribution, especially considering the non-conclusive finding of previous studies. Due to the number of details, it was probably misleading that we used a constant lidar ratio (values and uncertainties by Mattis et al. (2004) for central European haze aerosol) to derive the lidar-based aerosol light extinction. We agree that using the modeled LR for deriving extinction would be circular. We will emphasize that in the manuscript. We added in lines 334- 335: "...derived $\sigma_{\text{ext}}(\lambda)$. Later, the LR derived with the Mie model in the ambient state is compared with the LR provided by Mattis et al. (2004)."*

*Moreover, the shown results indicate that, especially for large RH ranges within the planetary boundary layer, it can be important to account for a humidity effect if the LR, and consequently the extinction, is enhanced by 2 to 3. Also, in the Fernald-Klett (Klett, 1985 and Fernald, 1984) algorithm, an inversion algorithm to derive aerosol optical properties from lidar, the lidar ratio is included, e.g., to estimate molecular extinction a certain altitude. The consideration of a relative humidity effect would thus contribute to an improvement.*

The overall conclusions and taken home messages of the study are completely lost in the in the details of the comparison. The goal of the study is the demonstrate the closure of aerosol optical property measurements, but fairly large differences remain. The authors speculate on several different reasons as to why the modeled and lidar-measured optical properties differ, but no real definitive answer is provided. The study would benefit from further analyses that are more focused on achieving closure to within a meaningful degree of certainty and a clear motivation/definition of what is "meaningful".

*Yes, to some extent, the significant differences remain. However, we only can provide some speculations of the reasons for the differences due to the interdependent underlying parameters. However, since we updated the motivation and title, the detailed analysis of various reasons is valuable for future studies.*

Before even attempting the closure exercise, it would be useful to discuss how good a closure one can expect given the uncertainties both the lidar and in-situ measurements. The uncertainties appear quite large at times which would suggest improved measurements techniques and methodologies are need before a useful closure exercise could be performed.

*At first, the uncertainty of the lidar system: Backscatter measurements were estimated with an assumed uncertainty of 10% following Wandinger et al. (2016). The aerosol particle light extinction derived from the lidar contains two uncertainty sources, the underlying backscatter measurements and the uncertainty of the LR used for the transformation. The latter was taken from Mattis et al. (2004) with given uncertainty estimates for central European haze aerosol. These are 58 ($\pm$12; 21%) sr for 355 nm, 53 ($\pm$11, 21%) sr for 532 nm, and 45 ($\pm$15, 33%) sr at 1064 nm. At best, the uncertainty of lidar is then:*

$$\delta\sigma_{\text{ext}} = \sqrt{\left(\frac{LR*\sigma_{\text{bsc}}}{dLR}*\Delta LR\right)^2 + \left(\frac{LR*\sigma_{\text{bsc}}}{d\sigma_{\text{bsc}}}*\Delta\sigma_{\text{bsc}}\right)^2} = \sqrt{(\sigma_{\text{bsc}}*\Delta LR)^2 + (LR*\Delta\sigma_{\text{bsc}})^2}.$$

*This equation translates into relative uncertainties of 23% at 355 nm, 23% at 532 nm, and 35% at 1064 nm. To provide uncertainties of the lidar extinction we updated the manuscript:*

*We added in lines 322- 323: "**LR is an intensive aerosol property**. The estimates of $\sigma_{ext}(\lambda)$ hence are subject of uncertainties arising from the LR uncertainty and the $\sigma_{ext}(\lambda)$ and $\sigma_{bsc}(\lambda)$.", and later in lines 333 - 337: "... with the LR provided by Mattis et al. (2004). With the uncertainty range of the LR by Mattis et al. (2004) and applying Gaussian error propagation, the uncertainty of the lidar-based $\sigma_{ext}(\lambda)$ is at best 23% at 355 nm, and 532 nm, and 35% at 1064 nm, respectively."*

*Uncertainties of the lidar measures are displayed with shaded areas around the vertical profiles and with error bars in the scatter plots.*

*Uncertainties of the modeled parameters are not as simple as for the lidar measurements. Like the density of volume fraction of eBC, many interdependent input parameters are fed into the model. Therefore, to some extent, we established the Monto-Carlo simulation to estimate the uncertainties of modeled parameters capturing the spatial and temporal variability of the input. So yes, with the used instrumentation and assumptions, a closure is hard to achieve. However, showing that is useful as well, especially if one plans a similar study in the future.*

Other comments:

The abstract needs to be considerably shorter. As is, it is a detailed summary of the entire paper and largely repetitive of the material in section 5.

*We rephrased the entire abstract:*

*" A unique data set derived from remote sensing, airborne, and ground-based in situ measurements is presented. The study highlights the complexity of comparing multiple aerosol optical parameters examined with different approaches considering different states of humidification and atmospheric aerosol concentrations. Mie-theory-based modeled aerosol optical properties are compared with respective results of airborne and ground-based in-situ measurements and remote sensing (lidar, photometer) performed at the rural central European observatory at Melpitz, Germany. Calculated extinction-to-backscatter ratios (lidar ratios) are in the range of previously reported values. However, the lidar ratio is not only a function of the prevailing aerosol type but also of the relative humidity. The particle lidar ratio (LR) dependence on relative humidity was quantified and followed the trend found in previous studies. We present a fit function for the lidar wavelengths of 355, 532, and 1064 nm with an underlying equation of fLR(RH, $\gamma(\lambda)$) = fLR(RH=0, $\lambda$)×(1-RH)-$\gamma(\lambda)$, with the derived estimates of $\gamma$(355 nm) = 0.29 (±0.01), $\gamma$(532 nm) = 0.48 (±0.01), and $\gamma$(1064 nm) = 0.31 (±0.01) for the central European aerosol. This parameterization might be used in the data analysis of elastic-backscatter lidar observations or lidar-ratio-based aerosol typing efforts. Our study shows that the used aerosol model was able to reproduce the in-situ measurements of the aerosol particle light extinction coefficients (measured at dry conditions) within 13%. Although the model reproduced the in situ measured aerosol particle light absorption coefficients within a reasonable range, we identified a number of sources for significant uncertainties in the simulations, such as the unknown aerosol mixing state, brown carbon (organic material) fraction, and the wavelength-dependent refractive index. The modeled ambient-state aerosol*

*particle light extinction and backscatter coefficients were found to be smaller than the measured ones. However, depending on the prevailing aerosol conditions, an overlap of the uncertainty ranges of both approaches was achieved."*

line 247: what "other studies"?

*Exemplarily, we added Höpner et al. (2016), Omar et al. (2009), Kim et al. (2018), and Rosati et al. (2016) and updated the references accordingly. The part is now:*

*"Therefore, in this and other studies, e.g., Omar et al. (2009), Kim et al. (2018), Rosati et al. (2016a), and Höpner et al. (2016), the $\sigma_{bsc}(\lambda)$ have been converted to $\sigma_{ext}(\lambda)$ utilizing the extinction-to-backscatter ratio, also known as lidar ratio (LR, in sr), with:"*

*Höpner, F., Bender, F. A.-M., Ekman, A. M. L., Praveen, P. S., Bosch, C., Ogren, J. A., Andersson, A., Gustafsson, Ö., and Ramanathan, V.: Vertical profiles of optical and microphysical particle properties above the northern Indian Ocean during CARDEX 2012, Atmos. Chem. Phys., 16, 1045–1064, https://doi.org/10.5194/acp-16-1045-2016, 2016.*

*Kim, M.-H., Omar, A. H., Tackett, J. L., Vaughan, M. A., Winker, D. M., Trepte, C. R., Hu, Y., Liu, Z., Poole, L. R., Pitts, M. C., Kar, J., and Magill, B. E.: The CALIPSO version 4 automated aerosol classification and lidar ratio selection algorithm, Atmos. Meas. Tech., 11, 6107–6135, https://doi.org/10.5194/amt-11-6107-2018, 2018.*

*Omar, A. H., Winker, D. M., Vaughan, M. A., Hu, Y., Trepte, C. R., Ferrare, R. A., Lee, K., Hostetler, C. A., Kittaka, C., Rogers, R. R., Kuehn, R. E., and Liu, Z.: The CALIPSO Automated Aerosol Classification and Lidar Ratio Selection Algorithm. J. Atmos. Ocean. Tech., 26, 10, 1994-2014, 2009.*

*Rosati, B., Herrmann, E., Bucci, S., Fierli, F., Cairo, F., Gysel, M., Tillmann, R., Größ, J., Gobbi, G. P., Di Liberto, L., Di Donfrancesco, G., Wiedensohler, A., Weingartner, E., Virtanen, A., Mentel, T. F., and Baltensperger, U.: Studying the vertical aerosol extinction coefficient by comparing in situ airborne data and elastic backscatter lidar, Atmos. Chem. Phys., 16, 4539–4554, https://doi.org/10.5194/acp-16-4539-2016, 2016a.*

line 357: add space after "campaign"

*Due to the rearrangement of the paper content, this part does not exist anymore.*

line 523: biasing -> attenuate

*We changed accordingly. Part is in the supplementary material now.*

line 863: remove "In Mie-theory"

*We changed as requested.*

Labels the panels in each figure (e.g. a, b, c) and use those labels when referring to specific panels in the text.

*We updated all Figures and corresponding references within the text accordingly.*

Figure 6, last panel: suggest addition a scale break in the x axis. As is, the x limits are too wide to discern any differences.

*We changed Figures 6 and 8 (now 3 and 5) accordingly by adding a break and squeezing the scale of the 2nd part (see exemplarily below).*

[Figure]

*Figure 1: Updated Figure 5.*

Figures 13, 14: the large amount of overlap in data points and errors bars make it difficult to see anything quantitative from these plots. These may be better plotted using some type of density-based plot.

*Thank for the comment. Since 1 Hz data are not providing any helpful information, we decided to reprocess the data of the STAP on a 30-second basis, which significantly reduced the noise. Moreover, we added a correction in terms of scattering and updated all estimates of depending parameters like AAE and the correlation. We plotted the lines thinner, and exemplarily the graphs for March 9, 2017, are shown below.*

[Figure]

*Figure 2: Updated Figure 8.*

[Figure]

*Figure 3: Updated Figure 10.*

---

## Author Comment (AC2)

Response to Referee #1:

*Blue italic font means the authors' answer,* standard, black *text symbolizes the reviewer's comments, and* red italic font *means manuscript changes, while* black italic *font symbolizes the original content of the manuscript.*

*Response:*

*We gratefully thank the anonymous referee for his comments and input. The mentioned points are addressed below.*

The article is still poorly written and tedious. Data from various measurements were used in this article including surface instrumentations like D-MPSS, Q-ACSM and lidar system. However, there are only lengthy analysis and few merits. Lidar has its advantage in measuring aerosol vertical hygroscopic growth in ambient environment, which is not evident in this article. In addition, errors due to the different observation methods and uncertainties from optical parameters from lidar should be considered.

*We shifted some parts into the supplementary to shorten the lengthy analysis. We updated the scope of the article to highlight some findings. For details on this modification, we refer to the answer to referee #3.*

*However, the scope of the article is not to show the lidar capability of deriving the aerosol hygroscopic growth (HG), although it would be helpful within this study to simulate the HG of the in-situ measured dry aerosol particles. However, comparing modeled optical parameters based on an HG simulation using lidar HG estimates would be somewhat circular. Nevertheless, the capability of lidar to inspect the hygroscopic behavior of aerosol is included in the introduction:*

*"Previous studies have focused on the dependence of $\sigma_{ext}(\lambda)$ on ambient RH (Skupin et al., 2013; Zieger et al., 2013). Navas-Guzmán et al. (2019) utilized these effects to investigate the aerosol hygroscopicity with lidar. $LR(\lambda)$ is based on the RH-dependent $\sigma_{bsc}(\lambda)$ and $\sigma_{ext}(\lambda)$, and calculations by Sugimoto et al. (2015) indicated that $LR(\lambda)$ is RH-dependent as well. Ackermann (1998) provided a numerical study based on pre-defined aerosol types with distinct size-distribution shapes to establish a power series to describe the $LR(\lambda)$ as a function of RH. Salemink et al. (1984) found a linear relationship between the $LR(\lambda)$ and the RH. Intensively discussed is the LR-enhancement due to hygroscopic growth in Zhao et al. (2017). They reported a positive relationship between LR and RH, but their study lacks information on vertically resolved aerosol particle number size distributions and other wavelengths. However, their simulations have shown that utilizing RH-dependent LR to retrieve aerosol particle light extinction from elastic backscatter lidar signals results in significantly different values than the constant LR approach."*

*Uncertainties of the lidar have been specified - 10% measurement uncertainty in terms of backscatter. Extinction estimates by the lidar are derived by LR provided by Mattis et al. (2004) and corresponding uncertainties. We also refer to the answer to referee #3, in which we described the uncertainty of the lidar-based extinction based on Gaussian error propagation.*

*Errors by the in-situ observations have been tackled utilizing a Monte-Carlo simulation.*

*Ongoing with the additional report of lidar-based studies investigating hygroscopic behavior of aerosol, we added Zhao et al. (2017) as an additional source for an LR(RH) parameterization and updated Figure 6 with the corresponding curve.*

[Figure]

*Figure 1: Updated Figure 6.*

*Zhao, G., Zhao, C., Kuang, Y., Tao, J., Tan, W., Bian, Y., Li, J., and Li, C.: Impact of aerosol hygroscopic growth on retrieving aerosol extinction coefficient profiles from elastic-backscatter lidar signals, Atmos. Chem. Phys., 17, 12133–12143, https://doi.org/10.5194/acp-17-12133-2017, 2017.*

The abstract is too long-winded and then innovation is difficult to find. Only one paragraph is suggested for the abstract.

*We updated the abstract as requested and refer to the answer to referee #3.*

Line 68: replace "does exsits" by "does exsit"

Line 101: replace "were" by "was"

Line 110: replace "is" by "are", Line 112: replace "is" by "are", Line 113: What is "This"?

*Due to structural changes in the manuscript these sentences do not exist anymore (see answer to referee #3).*

Line 128: delete "is"

*We cannot find anything wrong with the sentence: "Melpitz Observatory (51° 31' N, 12° 55' E; 84 m a.s.l.) is located in Eastern Germany in a rural, agriculturally used area 44 km northeast of Leipzig."*

Line 230: replace "underestimates" by "underestimate"

*Thanks for the suggestion. The sentence in line 294 is now: "However, the bulk Q-ACSM approach might over- or underestimates the hygroscopicity of aerosol particles lowersmaller or larger than 165 nm in diameter."*

Line 290: replace "of" by "for"

We changed accordingly. This part is shifted to the supplementary material.

Line 343: replace "were" by "was"

Since multiple filters were used within the study we changed "filter" to "filters".

Line 535: Poor writing

Thanks for the suggestion. We in the supplementary material lines 63- 65: " *Figure S2b) displays the time series of the number concentration of all aerosol particles up to a size of 800 nm in diameter.*"

Line 578: replace "led" to "lead"

We changed as requested.

Line 618: replace "oppoding to" by "opposed to"

We changed as requested: We changed in line 722 : " *Compared to* … "

Line 702: "solely based on…"

We changed as requested in line 665: "*…above 90% RH which we could not observe in this study*  *solely based on the small number of cases and the observed RH range.*" ()

---

## Author Comment (AC3)

Response to Referee #2:

*Blue italic font means the authors' answer,* standard, black *text symbolizes the reviewer's comments, and* red italic font *means manuscript changes, while* black *italic font symbolizes the original content of the manuscript.*

***Response:***

*We gratefully thank the referee for the spent effort and comments. We respond on the individual points below.*

This study compares lidar optical properties to those computed with Mie calculations in function of RH. The topic is important but the work suffers of important lacks in the method section probably biasing the obtained results and related considerations. Thus a deep major revision is required based on the following major issues:

Lines 290-292: "Also, the residual layer containing some aerosol layer aloft the top of the planetary boundary layer (PBL) between 1250 m and 2300 m is visible indicated by greenish colors." Given the description above and Figure 1 it is clear that ACTOS also sampled in the residual layer between ~1300 and ~2000m. I suggest to correct the sentence at line 292-293 ("The payload, therefore, was sampling in the free troposphere as well as within the planetary boundary layer and was sampling different aerosol populations") and ALL the related discussion and interpretation later in the results.

*Thanks for the comment. This part is transferred to the supplementary part of the manuscript. However, we added in line 34 - 35 in the supplementary material:* "The payload, therefore, was sampling in the free troposphere as well as within the planetary boundary layer and was sampling different aerosol populations."

*Of the four shown investigated flights of the summer campaign, two were conducted during a fully developed planetary boundary layer (flight 20150617b and flight 20150628b). Residual layers are observed for flights 20150626a and 20150628a (top right, and bottom left figure below). Below, the flight patterns of ACTOS during the measurement days are shown in the figures below. To raise the awareness of the audience, we added in line 482 - 490:* "The flight was conducted in the early morning from 08 to 10 UTC. During this daytime, the PBL is usually still developing due to thermal convection. Hence, most of the data were collected within the residual layer. The residual layer is an aged layer of aerosol, and the aerosol sampled on the ground should not represent the layer aloft the PBL. However, the model calculates aerosol particle light backscatter and extinction within 35% compared to the lidar with the best agreement at 532 nm, reproducing the extinction within 12%, much smaller than the approximated lidar uncertainty. Within the PBL, presumingly up to an altitude of 600 m, the model significantly calculates larger $\sigma_{ext}(\lambda)$ and $\sigma_{bsc}(\lambda)$. Surprisingly, the assumptions within the model capture the conditions within the residual layer better than the aerosol conditions within the PBL. Maybe the more aged aerosol within the residual fits better the core-shell mixing assumption within the model."

*And more in line 536 – 539:*

"Above the PBL, within the free troposphere, the model is significantly larger than the lidar estimates. However, ACTOS was not flying directly above the lidar; hence, small scale differences in the PBL height could explain the difference. These variations in the PBL height are also visible in Figure S1, with distinct variations of the aerosol load within a short period."

*However, the overall contribution to the total data set is small, and most of the data is collected within the PBL.*

[Figure]

*Figure 1: ACTOS flight track and attenuated backscatter coefficient measured by the Polly^XT lidar Arielle during the flight period of ACTOS on the measurement days of the summer campaign.*

Lines 296-297: how much below 40% RH the aerosol was sampled? Consider that aerosol efflorescence (or crystallization) can occur at RH lower than 40%, even below 30% RH in function of the aerosol chemical composition (nitrate to sulfate ratio, degree of acidity, presence of ammonium chloride etc…) (Martin, S. T.: Phase Transitions of Aqueous Atmospheric Particles., Chemical reviews, 100(9), 3403–3454, 2000). Please add a deep discussion based on this point as the manuscript aims at closure in function of RH, but the aforementioned consideration poses an important issues to the capability to reach this goal.

*We express many thanks for the comment. Although the efflorescence of hygroscopic aerosol particles is known, the effect is only observed in Melpitz during westerly inflows characterized by marine air masses, as Zieger et al. (2013) showed for multiple European sites, including Melpitz. During the winter campaign, between February 1 and March 15, 2017, a mean volume fraction of organic matter of 0.48 (median=0.74, IQR from 0.39 to 0.54) was observed, during the summer campaign period from June 1 to June 30, a mean volume fraction of 0.58 (median=0.59, IQR from 0.47 to 0.69). Due to these relatively high volume fractions, the hysteresis effect of scattering enhancement is not observed and therefore has not to be considered in the calculations.*

*We observed a maximum of 35.8% RH at all but one day downstream of the dryer. However, we will point out that the found parameterization is only applicable for non-marine air masses.*

*We added in lines 641 - 646: "Zieger et al. (2013) have shown the scattering enhancement due to hygroscopic growth for different European sites. In all but marine airmass-influenced cases, no hysteresis effect was observed at Melpitz, and they stated that these might occurs due to high fractions of low hygroscopic organic material. Hence, the effects of the aerosol efflorescence can be neglected since the volume fraction of the organic material within the aerosol population was relatively large during the summer campaign period. A mean volume fraction of 0.58*

*(median=0.59, IQR from 0.47 to 0.69) was estimated based on the chemical composition and assumed material densities within June 1 and June 30, 2015.", and in lines 682: "Nevertheless, the presented results provide good first estimates of the RH-induced LR(λ) enhancement factor based on in-situ measured PNSD for the observed RH range for the aerosol conditions at Melpitz. Although Ackermann (1998) ..."*

*Zieger, P., Fierz-Schmidhauser, R., Weingartner, E., and Baltensperger, U.: Effects of relative humidity on aerosol light scattering: results from different European sites, Atmos. Chem. Phys., 13, 10609–10631, https://doi.org/10.5194/acp-13-10609-2013, 2013.*

Lines 306-307, Figure 2 and Lines 316-321: The missing refractive index correction of the OPSS represents a lack of the manuscript in the way as it is actually presented. This section needs an improvement. For example, the inner "detailed geometry of the optical cell inside the instrument" should be asked to the manufacturer (or at least asking the equivalence with that reported in: Heim, M., Mullins, B. J., Umhauer, H., and Kasper, G.: Performance evaluation of three optical particle counters with an efficient "multimodal" calibration method, J. Aerosol Sci., 39, 1019–1031, doi:10.1016/j.jaerosci.2008.07.006, 2008).

*Thanks for the suggestion and input. Yes, Heim et al. (2008) provide insights on the geometry of the 1.109 optical particle sizer. Although with the given geometries, a Mie-based correction would be feasible, another reason prevents the correction of the refractive index. According to the manual, the GRIMM skyOPC is calibrated to a PSL calibrated mother device using polydisperse mineral dust (dolomite). This calibration was not reproducible within TROPOS.*

*Also, the results of Walser et al. (2017) indicate broad measurement spectra of mono-disperse PSL aerosols. These broad sizing spectra are not helpful to create a high valid refractive index correction. Moreover, for a refractive index correction, the polarization of the laser is needed but is unknown to our knowledge.*

*We updated the part in the manuscript as follows in line 407 - 410:" The manual of the skyOPC (v. 2.3) states that each offspring OPSS unit is calibrated to a mother instrument with an in-house standard using polydisperse mineral dust (dolomite). Walser et al. (2017) show broad sizing spectra of monodisperse polystyrene latex particle aerosols measured by the skyOPC. Also, the polarization of the used laser with a wavelength of 655 nm is unknown but is needed to calculate the response OPSS response curve.*  *Because of these reasons, a correction regarding the complex aerosol refractive index ($n = n_r + in_i$) could not be applied to the data set. The OPSS in-situ measurements were quality checked by comparing the average PNSD of the lowermost 200 m with the ground in-situ measurements (see Figure 2)."*

*Walser, A., Sauer, D., Spanu, A., Gasteiger, J., and Weinzierl, B.: On the parametrization of optical particle counter response including instrument-induced broadening of size spectra and a self-consistent evaluation of calibration measurements, Atmos. Meas. Tech., 10, 4341–4361, https://doi.org/10.5194/amt-10-4341-2017, 2017.*

Mie calculation should be biased using the OPSS optical equivalent diameters, thus affecting a part of section 3 (Modeling optical properties with Mie), discussion and all conclusions. The later (line 326) altitude correction factor in eq. 6 does not correct the OPSS optical equivalent aerosol size-bin (i.e. the size of particles) which is, instead, the right parameter needed for proper Mie calculations. It is required to clarify this point for the reader. Moreover, the above approach generate an inconsistency with lines 359-363 ("The OPSS PNSD was corrected in

terms of the complex aerosol refractive index. Here, a complex aerosol refractive index of 1.54 + i0 was used since this resulted in OPSS PNSD with a good overlap to the MPSS PNSD. The imaginary part of the complex aerosol refractive index was forced to 0 because it leads to a significant overestimation of the coarse mode in the PNSD when the imaginary part of the complex aerosol refractive index is above 0 (see Alas et al., 2019). Note, that this complex aerosol refractive index is not the refractive index used in the Mie model") and an inconsistency with lines 368-369 (Particles larger than 800 nm have not been replaced by the PNSD measurements at ground since the refractive index correction was applied to the OPSS data where different methods were used. I suggest to improve the discussion of the Mie methodology (and related approximations) from line till line 498 to make it clearer and more consistent.

*Thanks for the suggestion. First of all, to clarify and resolve some inconsistencies, the usage of the altitude correction factor is updated within the manuscript in lines 439 - 446:* "*In both cases, the instrumentation onboard the payloads did not cover the entire aerosol particle size range from 10 nm to 10 μm. Since the in-situ instrumentation at the ground is quality-assured, the ground-based measurements are the reference and are utilized to correct the airborne measurements. The missing size range is addressed as follows: The size range of the corresponding PNSD from the ground fills the missing size range; from 10 nm up to 326 nm, in the winter case, in the summer case, all sizes larger than 800 nm in optical diameter. Advantageously this addresses the unaccounted underestimation of larger particles by the skyOPC in the summer case and also provides volume-equivalent diameters for the Mie calculations in that size range. To account for vertical variability within the atmosphere, the ground-based PNSD is corrected for altitude, establishing a non-fixed altitude-correction factor $f_h$.*". $f_h$ *corrects the missing part of the airborne PNSD and also the number concentration of particles, which is also needed to proper model aerosol optical properties.*

*In the case of the summer, yes, the uncorrected OPSS size distribution biases the Mie model results. The estimate of the bias induced by not correcting the skyOPC PNSD in this size range is challenging and cannot be determined easily. However, the airborne extinction is reproduced by the model sufficiently. In the analysis of the data, we also tackled that issue in lines 556-558:* "*Moreover, as the refractive index correction of OPSS tends to shift the particle towards a larger diameter, at least partially, that could explain some of the underestimations, although the used size range of the skyOPC is limited between 356 and 800 nm.*". *Moreover, as clarification, we added in line 428 - 429:* " *Contrary to the PNSD derived with the sykOPC, this OPSS PNSD is corrected with in-house software in terms of the complex aerosol refractive index.*" *We assume a small impact since the uncorrected sykOPC mean PNSD of the lowermost 200m (Figure 2) is partially smaller than and partially larger than the PNSD at derived ground level.*

*In the winter campaign, the OPSS is corrected with a refractive index of 1.54, and calculations with 1.56 can not explain the difference of both approaches (see lines 855 - 857).* "*However, using the ZSR-based real part of the complex refractive index of 1.56 during both days cannot explain the lidar and Mie model differences. Applying this real part to the data of February 9, the slope of the correlation changes within absolute values of -0.055 to 0.045 compared to a real part of 1.54.*"

*Biased refractive indices for the TSI OPSS correction have been addressed in the manuscript in lines 433 - 438:*" *For the investigated days of the winter campaign, a median complex refractive index of the aerosol of 1.56+i0.11 is found for February 9 and 1.56+i0.06 for March*

*9, respectively. However, these refractive indices are based on the ZSR mixing of homogeneously mixed particles but, a) we assumed a core-shell mixing of the aerosol particles and b) the shape of the aerosol particles is essential as well for the refractive index correction. Therefore, the used complex refractive index for correction is more an effective refractive index to match the OPSS PNSD to the PNSD derived at ground level with the MPSS and APSS."*

Lines 350-351: "truncation error of the scattering coefficient was not corrected". Please, add also the uncertainty of scattering and not only that of extinction.

*Thanks for the suggestion. The estimation of the scattering uncertainty depends on the truncation and calibration error of the CAPS. The truncation error depends on the aerosol morphology, the aerosol particle number size distribution, and the aerosol refractive index.*

*Within the CAPS a PMT (photomultiplier tube) including an integrating sphere is installed to measure the scattered light. Integrating sphere nephelometers like the one used in this study at the ground station measure the scattered light similarly and the measurement uncertainty due to calibration and truncation is usually not more than 10%. See line 203 – 205: "These measurements were completed by a Nephelometer (mod. 3563, TSI Inc., Shoreview, MN, USA), which measures the $\sigma_{sca}(\lambda)$ at 450, 550, and 700 nm with a relative uncertainty by calibration and truncation of about 10% (Müller et al., 2009)."*

*Since the airborne measured aerosol particle light scattering coefficient was not used within this study, there is no point in providing the measuring uncertainty. However, Modini et al. (2021) recently provided a detailed characterization of the CAPS PMssa instrument. Truncation correction factors were provided with uncertainties of 4% and 9% for fine and coarse mode dominated aerosols, respectively.*

*We added to the manuscript in lines 391 - 395: "The measured aerosol particle light scattering coefficient is not used within this study and therefore he truncation error of $\sigma_{sca}(630\ nm)$ is not corrected. Moreover,  we focus on $\sigma_{ext}(630\ nm)$ estimated with a 5% accuracy. However, a detailed characterization of the CAPS PMssa monitor is provided by Modini et a. (2021). Truncation and scattering cross-calibration correction factors were provided with uncertainties of 2%, and 4% to 9% for fine and coarse mode dominated aerosol, respectively."*

*Modini, R. L., Corbin, J. C., Brem, B. T., Irwin, M., Bertò, M., Pileci, R. E., Fetfatzis, P., Eleftheriadis, K., Henzing, B., Moerman, M. M., Liu, F., Müller, T., and Gysel-Beer, M.: Detailed characterization of the CAPS single-scattering albedo monitor (CAPS PMssa) as a field-deployable instrument for measuring aerosol light absorption with the extinction-minus-scattering method, Atmos. Meas. Tech., 14, 819–851, https://doi.org/10.5194/amt-14-819-2021, 2021.*

Lines 360-363: OPSS model 3330 of TSI only accept real part of refractive index. The use of 1.54 + i0 is mandatory, not a decision. Moreover, this can generate problems if "this complex aerosol refractive index is not the refractive index used in the Mie model" as reported. Please comment and clarify.

*We have to clarify that to correct the optical diameters of the OPSS, we used in-house software in which the real part and imaginary part of the complex refractive index can be varied.*

*During the intensive period between February 1 and March 16, the median real part of the aerosol particles was 1.558, the imaginary part 0.08. On both investigated days, a mean*

*complex refractive index of 1.56+i0.109 on February 9 was observed, 1.56+i0.06 on March 9, respectively. The issue is addressed in the comments above.*

---

## Referee Report (RR1)

I appreciate all the revisions. But I still think the manuscript still failed to meet the requirement of publication in ACP. Firstly, although better than previous version, the manuscript was still written poorly and should further undergo extensive English revisions. Many measurement platforms were used in this article such as ground in-situ measurements, ground lidar and airborne in-situ measurements to present the comparison of lidar retrieved optical properties with airborne in-situ measurement-based ones, but the details in the comparison are very rough, which might lead to lots of errors in either the calculation of model optical properties or the lidar retrieved optical properties. How can they be comparable? Does the comparison make sense? Besides, I would suggest authors to address other major weaknesses, which are listed below.

1) Line 15: In the abstract, the author use "The study highlights the complexity of …", which is a well-known issue and is what we really want to solve. Through the whole manuscript, I can only find the author highlight the complexity without an actual solution, and there are few innovations for both results and the method during the comparison process.

2) Line 300: replace "2.1.2 Ground-based remote sensing" by "3.1.2 Ground-based remote sensing"

3) Line 316: "During the daytime, the signal-to-noise ratio in the Raman channels is too weak …" and the author use constant LR to retrieve aerosol optical properties, which will lead to huge errors especially for multi-wavelength lidar, different observation sites and experiment dates. The lidar data at night are free of the noise problem. Why not try to using these data to calculate the LR?

4) Line 344: replace "2.1.3 Airborne in-situ measurements" by "3.1.3 Airborne in-situ measurements"

5) Line 479-480: "… and slightly lower than…That indicates different aerosol populations in these layers" These might also result from errors during the calculation, such as the determination of refractive index and lidar ratio and the particle size distribution range used in Mie model, which can't be ignored and determine whether

the comparison results were meaningful.

6) Line 460, 515: The figure c), d) and e) are not easily readable. Some legends can't be found in the figure.

7) Line 515: The same as Line 479-480.

8) Line 627- 630: How long did the flights 20150617b, 0626a, 0628a, 0628b last? Which heights and locations were chosen for the LR calculation? If there the particle size distributions were influenced by air mass transported from other regions, how can you guarantee the changes in calculated LR were merely resulting from the relative humidity?

9) Line 929 - 942: Is the conclusion here necessary?

---

## Author Response (AR2)

Response to Referee #1:

*Blue italic font means the authors' answer,* standard, *black* *text symbolizes the reviewer's comments,* *and* *red italic* font *means manuscript changes, while* black *italic* *font symbolizes the original content of* *the manuscript.*

I appreciate all the revisions. But I still think the manuscript still failed to meet the requirement of publication in ACP. Firstly, although better than previous version, the manuscript was still written poorly and should further undergo extensive English revisions. Many measurement platforms were used in this article such as ground in-situ measurements, ground lidar and airborne in-situ measurements to present the comparison of lidar retrieved optical properties with airborne in-situ measurement based ones, but the details in the comparison are very rough, which might lead to lots of errors in either the calculation of model optical properties or the lidar retrieved optical properties. How can they be comparable? Does the comparison make sense? Besides, I would suggest authors to address other major weaknesses, which are listed below.

*Response:*

*We gratefully thank the anonymous referee for his comments and input. The mentioned points are addressed below. We updated lengthy sentences and revised them in terms of grammar.*

1) Line 15: In the abstract, the author use "The study highlights the complexity of …", which is a well-known issue and is what we really want to solve. Through the whole manuscript, I can only find the author highlight the complexity without an actual solution, and there are few innovations for both results and the method during the comparison process.

*Since the manuscript type is changed to "measurement report", we argue that providing a definite answer to resolve the uncertainty is not required for the scope of this journal's manuscript type. Also, due to the complexity of aerosol and missing information of the atmospheric column, no definite answer can be given. Nevertheless, the data set and manuscript are valuable sources for planning future campaigns and highlight, e.g., the know dependence of the LR to RH or addressing the mentioned issues by selecting appropriate instrumentation and shows demand for new instrumentation that can resolve these issues.*

2) Line 300: replace "2.1.2 Ground-based remote sensing" by "3.1.2 Ground-based remote sensing

*We updated as requested.*

3) Line 316: "During the daytime, the signal-to-noise ratio in the Raman channels is too weak …" and the author use constant LR to retrieve aerosol optical properties, which will lead to huge errors especially for multi-wavelength lidar, different observation sites and experiment dates. The lidar data at night are free of the noise problem. Why not try to using these data to calculate the LR?

*Yes, determining the LR during the night could have been a method to derive the LR for the column. However, at 1064 nm, the lidar ratio could not be measured at this time (first approaches have just evolved, e.g., Haarig et al., 2018); thus, we used corresponding values provided in the literature (45 ± 15 sr; Mattis et al., 2004).*

*Furthermore, if possible, the utilized lidar ratios have been validated by comparing the height-integral of the aerosol particle light extinction coefficient at 532 and 355 nm with the corresponding AOD measured by the deployed sun-photometer. However, for June 26 and 17, 2015, no sun-photometer data is available due to cloud coverage during these days. For June 28, 2015, the integral of the mean aerosol particle light coefficient between 0 and 2500 m and 8 to 10 UTC (below the overlap height, the values are linearly extrapolated to the ground) is 0.13 at 355 nm and 0.072 at 532 nm. The corresponding AOD(355 nm), extrapolated with the Angstrom between 340 and 380 nm, is 0.14 and 0.097 at 532 nm (extrapolated between 500 and 675 nm). However, since no sun-photometer data for validation is available for June 26, and 17 and the lidar ratios provided by Mattis et al. (2004) seem to fit reasonably well for June 28, 2015, we decided to stick with the lidar ratios provided by Mattis et al. (2004).*

*Hence, we added (342 - 349):* "*Directly deriving the LR from nighttime observations with the Raman-Lidar would also have been a feasible approach. However, as the atmospheric conditions between night and daytime were not homogenous and quite variable, we could not apply the nighttime finding to our daytime observations. However, we used AERONET AOD data to validate our extinction profiles and found good agreement whenever atmospheric conditions allowed. E.g., for June 28, 2015, the integral of the mean aerosol particle light coefficient between 0 and 2500 m and 8 to 10 UTC (below the overlap height, the values are linearly extrapolated to the ground) is 0.13 at 355 nm and 0.072 at 532 nm. The corresponding AOD(355 nm), extrapolated with the Ångström exponent between 340 and 380 nm, is 0.14 and 0.097 at 532 nm (extrapolated between 500 and 675 nm). Thus, we believe the used lidar ratio values are well justified.*"

*References: Haarig, M., Ansmann, A., Baars, H., Jimenez, C., Veselovskii, I., Engelmann, R., and Althausen, D.: Depolarization and lidar ratios at 355, 532, and 1064 nm and microphysical properties of aged tropospheric and stratospheric Canadian wildfire smoke, Atmos. Chem. Phys., 18, 11847–11861, https://doi.org/10.5194/acp-18-11847-2018, 2018.*

4) Line 344: replace "2.1.3 Airborne in-situ measurements" by "3.1.3 Airborne in-situ measurements"
*We updated as requested.*

5) Line 479-480: "… and slightly lower than… That indicates different aerosol populations in these layers" These might also result from errors during the calculation, such as the determination of

refractive index and lidar ratio and the particle size distribution range used in Mie model, which can't be ignored and determine whether the comparison results were meaningful.

75 *Yes, the different aerosol population is part of the uncertainty. However, based on the measurement setup, some important aerosol properties, e.g., the refractive index, could not be determined within the atmospheric column. As a result, the considerable uncertainty in the LR should cover a broad range to cover changes in aerosol type. Also, the lidar ratio can be ignored when comparing the backscatter coefficient, and only uncertainties in the aerosol particle number size distribution and chemical*

80 *composition impact the modeled values. Hence, the measurement and comparison are meaningful since the report highlights the challenges of vertical in-situ measurement-based studies with these considerable uncertainties.*

6) Line 460, 515: The figure c), d) and e) are not easily readable. Some legends can't be found in the

85 figure. And 7) Line 515: The same as Line 479-480.

*We updated the figures and added corresponding legends in panel a) and b). Accordingly, we updated Figures 7 and 8.*

[Figure]

**Figure 1: New Figure 8.**

[Figure]

90

**Figure 2:New Figure 7.**

[Figure]

**Figure 3: New Figure 5.**

[Figure]

Figure 4:New Figure 3.

8) Line 627- 630: How long did the flights 20150617b, 0626a, 0628a, 0628b last? Which heights and locations were chosen for the LR calculation? If there the particle size distributions were influenced by air mass transported from other regions, how can you guarantee the changes in calculated LR were merely resulting from the relative humidity?

*Typical flight duration and altitude coverage are stated in the Supplementary Material. All flights were planned to last around 2 hours and were conducted up to altitudes of 2700 m. In detail, flight 20150617b lasted from 12:43 to 14:19 UTC, 20150626a from 08:08 to 09:58 UTC, 20150628a from 09:07 to 11:04 UTC, and 20150628b form 13:05 to 14:48 UTC. The exact flight time for flights 20150626a and 20150617b is stated in lines 471 and 518.*

*All data points from the four selected flights are part of the LR calculation. For this, each available aerosol particle number size distribution is taken within and above the PBL. In particular, the altitude range of flight 20150617b is 0 to 2280 m, 20150626a, 0 to 2660 m, 20150628a 0 to 2380 m and 20150628b 0 to 2670 m. Hence, we agree that we cannot guarantee that the increase in relative humidity solely causes the increase of the aerosol particle light extinction coefficient when we talk about the vertical profile of the LR.*

*However, Figure 5 displays the profile of the LR in the ambient and humidified state and the RH profile of the given flights. The profile of the RH influences the LR more than the aerosol population indicated by relatively constant LR with height at the dry state. Nevertheless, the vertical aerosol distribution is not necessary for the investigation of the LR-RH-behavior because the LR-RH-dependence was calculated for each particle number size distribution itself. At most, the type of aerosol can influence the fitting exponent γ(λ) of the LR-RH-fit and is already discussed in the manuscript (lines 655 – 658, manuscript after first round of revision).*

[Figure]

**Figure 5: Profile of *RH* (A) and lidar ratio modeled in dry (circles) and ambient state (triangles) for the four investigated flights of the study.**

*However, to clarify the direct relationship between the LR and RH and to examine the influence of the dry state aerosol, we added (line 635 - 637): "Aerosol changes with height probably cause some changes in the LR too. However, a comparison of the LR profile in the dry state with the LR profile in the ambient state shows that the LR increases more with increasing RH than it does with a change in the aerosol itself (see Figure S7).", and referred therein to a new section of the supplementary material:*

**LR at dry state vs. ambient state**

[Figure]

*Figure S7: Profile of RH (panel A) and lidar ratio (panel B) modeled in dry (circles) and ambient state (triangles) for the four investigated flights of the study.*

*Figure S6 displays the profile of RH (panel A) measured during the four flights of interest of the*
130 *summer campaign. Panel B of Figure S7 shows the profiles of the LR at 355, 532, and 1064 nm calculated with the Mie model in ambient (triangles) and dry (circles) states. While the LR changes only slightly, if at all, in the dry state, the change is more pronounced in the ambient state. Hence a direct relationship between LR and RH is visible.*

135 9) Line 929 - 942: Is the conclusion here necessary?

*Yes, we think it is a neat way to summarize the conclusion. Some articles have been presented with that kind of summary, and since the first reviewer does not criticize, we keep it as it is. However, we shortened the conclusion and combined point a and b: It states now:*

*"       In conclusion:*

140 *a)       Conducting comparison studies of aerosol optical properties, e.g., to validate lidar-based $\sigma_{abs}(\lambda)$, requires a precise determination of the aerosol mixing state, its composition, the inclusion of BrC, and the application of a wavelength-dependent complex refractive index. Information on size- and height-resolved aerosol composition is needed.*

*b)       Airborne in-situ measurements of, e.g., the aerosol chemical composition, including the BrC*
145 *content, would improve studies focusing on the validation of lidar-based $\sigma_{abs}(\lambda)$.*

*c)b)       A wide range of aerosol particle sizes is covered in this study. However, the modeled $\sigma_{bsc}(\lambda)$ were on average lower than the measured one. A much further extension of the observedObserving*

*aerosol particles above a size  of 10 μm would ensure that these non-observed particles would not cause a significant  bias based on De Leeuw and Lamberts (1987).*

*c)     Knowing the connection between RH and the LR(λ), the LR(λ) enhancement can be a valuable tool to estimate the LR(λ) at ambient state when the dry state LR(λ) is known. Also, it allows calculating back the LR(λ) in the dry state, when the LR(λ) is directly measured in the ambient state, and an RH profile is known, e.g., via radio soundings.*

*d)     However, long-term measurements must be conducted to verify the LR(λ) enhancement estimates for various aerosol-types and different seasons."*

Response to Referee #2:

*Blue italic font means the authors' answer,* standard, black *text symbolizes the reviewer's comments, and* **red italic font** *means manuscript changes, while* **black italic** *font symbolizes the original content of the manuscript.*

The manuscript was improved and I thank the authors for the effort done. However, in my opinion, before publication, a further improvement is required concerning 3 aspects:

***Response:***

*We gratefully thank the referee for the spent effort and comments. We respond to the individual points below.*

1) Concerning my question on aerosol drying and the hysteresis cycle, the given answer (based on Zieger et al. 2013) is not convincing. Other measurements in Europe, in high organic mass fraction conditions (not marine aerosol) showed aerosol hysteresis (Fierz-Schmidhauser et al., 2010; Ferrero et al., 2019). However, I understand that the used apparatus cannot solve the question as at lines 639-640 is reported: "The RH measured after the dryer was at most 48.3% on flight 20150617b and reached a maximum of 35.8% on the other days". Thus my suggestion is to add a comment concerning the uncertainty expected in case in which the aerosol was not completely dried by the used apparatus.

References:

R. Fierz-Schmidhauser, P. Zieger, G. Wehrle, A. Jefferson, J. A. Ogren, U. Baltensperger, and E. Weingartner. Measurement of relative humidity dependent light scattering of aerosols. Atmos. Meas. Tech., 3, 39–50, 2010 www.atmos-meas-tech.net/3/39/2010/

Ferrero, L., Riccio, A., Ferrini, B.S., D'Angelo, L., Rovelli, G., Casati, M., Angelini, F., Barnaba, F., Gobbi, G.P., Cataldi, M., Bolzacchini, E., 2019. Satellite AOD conversion into ground PM10, PM2.5 and PM1 over the Po valley (Milan, Italy) exploiting information on aerosol vertical profiles, chemistry, hygroscopicity and meteorology. Atmospheric Pollution Research 10, 1895–1912. doi:10.1016/j.apr.2019.08.003.

*Figure 6 displays the RH post the deployed dryer. During 20150617b the RH was exceeding the threshold RH of 40% just occasionally. After that measurement flight, the dryer was replaced by a new one, ensuring the drying capability. However, backward-simulations of the hygroscopic growth show that the difference in diameter at 48% is at most 3.2% compared to 35% RH (kappa =0.3, temperature = 25°C), which is the maximum RH reached during flight 20150626a. Although the scattering, backscattering, and absorption efficiency change with change in particle diameter, the resulting optical coefficients are proportional to the cross-section of the particles, which is proportional to the square of the diameter. Hence in the "above 40% RH" case, the deviation is in this regard 6.5%. Compared to 40% RH, the "dry-state" threshold, the deviation is 2.1% or 4.2%, respectively.*

[Figure]

**Figure 6: Post dryer RH of measurement flight 20150617b (red) and 20150626a (black).**

*We added the comment:" In the 48%-RH case, the difference in RH results in a deviation of 3.2% in dry state diameter. The optical coefficients from the Mie calculation are proportional to the cross-section of the aerosol particle. Hence, the dry-diameter deviation translates into a deviation of 6.5% in this regard.".*

2) Concerning my question on OPSS PNSD corrections two points in the given answer needs to be clarified:

Point A) at lines 428-429 of the revised manuscript it is reported that "this OPSS PNSD is corrected with in-house software in terms of the complex aerosol refractive index". The development of an in-house software for OPSS correction in terms of the complex aerosol refractive index is yet a matter of a research paper. Here it is used with the aim to correct the size distribution to perform then a closure experiment. Thus there is the need to carefully describe the software and the related performance together with the inner geometry of the OPSS used for the correction. I strongly reccomend to add this part to the supplemental material.

*We agree that the correction of OPSS PNSD is still challenging since, besides the refractive index; also the aerosol particles' shape and mixing state are essential. As a best guess, the underlying software calculates with Mie theory, based on the geometry and features of the device, the intensity of the sideward scattered light for particles of a given refractive index. In detail: The software adjusts the particle diameter that the intensity of its sideward scattered light matches the intensity of the calibrated PSL particles. Hence for each size bin, the bin borders are shifted and adjusted to the new refractive index. For the TSI OPSS (mod. 3330), the opening angle of the aperture of 60° (±60°) is used. The wavelength of the device is 660 nm, and the refractive index of the PSL particles is set to 1.581+i0.*

*Calculations are conducted under the assumption of unpolarized light. The software was used already in Alas et al. (2019).*

*We added (line 438 - 444): "Briefly, the used software utilizes Mie theory to calculate the intensity of sideward scattered light with a given wavelength of aerosol particles with a complex refractive index and a given diameter D within an angular range. The next step shifts the diameter up to the intensity that matches the intensity of the calibration aerosol (here PSL) of a specific diameter and refractive index. As a result, the size bins are remapped to a new diameter array. For the calculations, the specific characteristics of the device have to be known. In this case, the sideward angular range is ±60°, the wavelength is 660 nm assuming unpolarized light and a refractive index of the calibration aerosol at this wavelength of 1.581+i0."*

Point B) as asked, "the altitude correction factor (fh) in eq. 6 does not correct the OPSS optical equivalent aerosol size-bin (i.e. the size of particles) which is, instead, the right parameter needed for proper Mie calculations". The answer does not shed a light on fh, probably it was a my fault to present a vague question. Eq 6 reports fh=N_OPSS(h)/N_OPSS(<x m), where N_OPSS is the number concentration. Thus it is necessary to clarify if the authors are using the total number concentration, the one from each size bine, or a normalized one just to correct the size distribution shape with altitude

*In the summer campaign, the PNSD measured by the OPSS was not corrected because of the presented reasons. In the winter campaign, we used in-house software to do so. We lacked information on non-observed size ranges (summer from 800 nm on; winter, up to 300 nm). Hence, these missing parts were corrected by the corresponding PNSD measured at ground level normalizing its shape (each size bin with the same factor) with the ratio of the aerosol particle number concentration detected close to the ground and the one detected at higher altitudes.*

*The respective part (line 462 - 464) states now: This factor normalizes the ground-based PSND (each bin equally) with the number concentration ratio of the aerosol particles detected by the OPSS at altitude h ($N_{OPSS}(h)$) and the mean in a layer near ground below an altitude x ($N_{OPSS}(<x\ m)$). The altitude-correction factor $f_h(h)$ is calculated according to Eq. (8):*

$$f_h(h) = \frac{N_{OPSS}(h)}{N_{OPSS}(<x\ m)}. \tag{8}$$

.

3) The revised version of the manuscript presents at line 151 equation 8 as the first equation. It is necessary to carefully check all the paper, the equation numbers and the related reference along the text

*We updated as requested. Changes are highlighted in the marked-up version.*